# State Chrono Representation for Enhancing Generalization in Reinforcement Learning

**Jianda Chen**[1]    **Wen Zheng Terence Ng**[1,2]    **Zichen Chen**[3]
**Sinno Jialin Pan**[4]    **Tianwei Zhang**[1]
[1]Nanyang Technological University    [2]Continental Automotive Singapore
[3]University of California, Santa Barbara    [4]The Chinese University of Hong Kong
{jianda001, ngwe0099}@ntu.edu.sg    zichen_chen@ucsb.edu
sinnopan@cuhk.edu.hk    tianwei.zhang@ntu.edu.sg

## Abstract

In reinforcement learning with image-based inputs, it is crucial to establish a robust and generalizable state representation. Recent advancements in metric learning, such as deep bisimulation metric approaches, have shown promising results in learning structured low-dimensional representation space from pixel observations, where the distance between states is measured based on task-relevant features. However, these approaches face challenges in demanding generalization tasks and scenarios with non-informative rewards. This is because they fail to capture sufficient long-term information in the learned representations. To address these challenges, we propose a novel State Chrono Representation (SCR) approach. SCR augments state metric-based representations by incorporating extensive temporal information into the update step of bisimulation metric learning. It learns state distances within a temporal framework that considers both future dynamics and cumulative rewards over current and long-term future states. Our learning strategy effectively incorporates future behavioral information into the representation space without introducing a significant number of additional parameters for modeling dynamics. Extensive experiments conducted in DeepMind Control and Meta-World environments demonstrate that SCR achieves better performance comparing to other recent metric-based methods in demanding generalization tasks. The codes of SCR are available in https://github.com/jianda-chen/SCR.

## 1 Introduction

In deep reinforcement learning (Deep RL), one of the key challenges is to derive an optimal policy from highly dimensional environmental observations, particularly images [5, 12, 33]. The stream of images received by an RL agent contains temporal relationships and significant spatial redundancy. This redundancy, along with potentially distracting visual inputs, poses difficulties for the agent in learning optimal policies. Numerous studies have highlighted the importance of building state representations that can effectively distinguish task-relevant information from task-irrelevant surroundings [45, 46, 8]. Such representations have the potential to facilitate the RL process and improve the generalizability of learned policies. As a result, representation learning has emerged as a fundamental aspect in the advancement of Deep RL algorithms, gaining increased attention within the RL community [21].

**Related Works.** The main objective of representation learning in RL is to develop a mapping function that transforms high-dimensional observations into low-dimensional embeddings. This process helps reduce the influence of redundant signals and simplify the policy learning process. Previous research has utilized autoencoder-like reconstruction losses [43, 18], which have shown

38th Conference on Neural Information Processing Systems (NeurIPS 2024).

impressive results in various visual RL tasks. However, due to the reconstruction of all visual input including noise, these approaches often lead to overfitting on irrelevant environmental signals. Data augmentation methods [42, 22, 23] have demonstrated promise in tasks involving noisy observations. These methods primarily enhance perception models without directly impacting the policy within the Markov Decision Process (MDP) framework. Other approaches involve learning auxiliary tasks [33], where the objective is to predict additional tasks related to the environment using the learned representation as input. However, these auxiliary tasks are often designed independently of the primary RL objective, which can potentially limit their effectiveness in improving the overall RL performance.

In recent advancements, behavioral metrics, such as the bisimulation metric [11, 10, 6, 4, 5, 47] and the MICo distance [7], have been developed to measure the dissimilarity between two states by considering differences in immediate reward signals and the divergence of next-state distributions. Behavioral metrics-based methods establish approximate metrics within the representation space, preserving the behavioral similarities among states. This means that state representations are constrained within a structured metric space, wherein each state is positioned or clustered relative to others based on their behavioral distances. Moreover, behavioral metrics have been proven to set an upper bound on state-value discrepancies between corresponding states. By learning behavioral metrics within representations, these methods selectively retain task-relevant features essential for achieving the optimal policy, which involves maximizing the value function and shaping agent behaviors. At the same time, they filter out noise that is unrelated to state values and behavioral metrics, thereby improving robustness in noisy environments.

However, behavioral metrics face challenges when handling demanding generalizable RL tasks and scenarios with sparse rewards [20]. While they can capture long-term behavioral metrics to some extent through the temporal-difference update mechanism, their reliance on one-step transition data limits their learning efficiency, especially in the case of sparse rewards. This limitation may result in representations that encode non-informative signals [20] and can restrict their effectiveness in capturing long-term reward information. Some model-based approaches attempt to mitigate these issues by learning transition models [14, 16, 27]. However, learning a large transition model with long trajectories requires a large number of parameters and significant computational resources. Alternatively, $N$-step reinforcement learning methods can be used to address the problem [48]. Nevertheless, these methods introduce higher variance in value estimation compared to one-step methods, which may lead to unstable training.

**Contributions.** To overcome the above challenges, we introduce the State Chrono Representation (SCR) learning framework. This metric-based approach enables the learning of long-term behavioral representations and accumulated rewards that span from present to future states. Within the SCR framework, we propose the training of two distinct state encoders. One encoder focuses on crafting a state representation for individual states, while the other specializes in generating a *Chronological Embedding* that captures the relationship between a state and its future states. In addition to learning the conventional behavioral metric for state representations, we introduce a novel behavioral metric specifically designed for temporal state pairs. This new metric is approximated within the chronological embedding space. To efficiently approximate this behavioral metric in a lower-dimensional vector space, we propose an alternative distance metric that differs from the typical $L_p$ norm. To incorporate long-term rewards information into these representations, we introduce a "measurement" that quantifies the sum of rewards between the current and future states. Instead of directly regressing this measurement, we impose two constraints to restrict its range and value. Note that SCR is a robust representation learning methodology that can be integrated into any existing RL algorithm.

In summary, our contributions are threefold: 1) We introduce a new framework, SCR, for representation learning with a focus on behavioral metrics involving temporal state pairs. We also provide a practical method for approximating these metrics; 2) We develop a novel measurement specifically adapted for temporal state pairs and propose learning algorithms that incorporate this measurement while enforcing two constraints; 3) Through experiments conducted on the Distracting DeepMind Control Suite [34], we demonstrate that our proposed representation exhibits enhanced generalization and efficiency in challenging generalization tasks.

## 2 Preliminary

**Markov Decision Process:** A Markov Decision Process (MDP) is a mathematical framework defined by the tuple $\mathcal{M} = (\mathcal{S}, \mathcal{A}, P, r, \gamma)$. Here $\mathcal{S}$ represents the state space, which encompasses all possible states. $\mathcal{A}$ denotes the action space, comprising all possible actions that an agent can take in each state. The state transition probability function, $P$, defines the probability of transitioning from the current state $s_t \in \mathcal{S}$ to any subsequent state $s_{t+1} \in \mathcal{S}$ at time $t$. Given the action $a_t \in \mathcal{A}$ taken, the probability is denoted as $P(s_{t+1}|s_t, a_t)$. The reward function, $r : \mathcal{S} \times \mathcal{A} \to \mathbb{R}$, assigns an immediate reward $r(s_t, a_t)$ to the agent for taking action $a_t$ in state $s_t$. The discount factor, $\gamma \in [0, 1]$, determines the present value of future rewards.

A policy, $\pi : S \times A$, is a strategy that specifies the probability $\pi(a|s)$ of taking action $a$ in state $s$. The objective in an MDP is to find the optimal policy $\pi^*$ that maximizes the expected discounted cumulative reward, $\pi^* = \arg\max_\pi \mathbb{E}_{a \sim \pi}[\sum_t \gamma^t r(s_t, a_t)]$.

**Behavioral Metric:** In DBC [47], the bisimulation metric defines a pseudometric $d : \mathcal{S} \times \mathcal{S} \to \mathbb{R}$ to measure the distance between two states. Additionally, a variant of bisimulation metric, known as $\pi$-bisimulation metric, is defined with respect to a specific policy $\pi$.

**Theorem 2.1** ($\pi$-bisimulation metric)**.** *The $\pi$-bisimulation metric update operator $\mathcal{F}_{bisim} : \mathbb{M} \to \mathbb{M}$ is defined as*

$$\mathcal{F}_{bisim}d(\mathbf{x}, \mathbf{y}) := |r_\mathbf{x}^\pi - r_\mathbf{y}^\pi| + \gamma \mathcal{W}(d)(P_\mathbf{x}^\pi, P_\mathbf{y}^\pi),$$

*where $\mathbb{M}$ is the space of $d$, $r_\mathbf{x}^\pi = \sum_{a \in \mathcal{A}} \pi(a|\mathbf{x})r_\mathbf{x}^a$, $P_\mathbf{x}^\pi = \sum_{a \in \mathcal{A}} \pi(a|\mathbf{x})P_\mathbf{x}^a$, $\mathcal{W}$ is the Wasserstein distance, and $r_\mathbf{x}^a$ is $r(\mathbf{x}, a)$ for short. $\mathcal{F}_{bisim}$ has a unique least fixed point $d_{bisim}^\pi$.*

MICo [7] defines an alternative metric that samples the next states instead of directly measuring the intractable Wasserstein distance.

**Theorem 2.2** (MICo distance)**.** *The MICo distance update operator $\mathcal{F}_{MICo} : \mathbb{M} \to \mathbb{M}$ is defined as*

$$\mathcal{F}_{MICo}d(\mathbf{x}, \mathbf{y}) := |r_\mathbf{x}^\pi - r_\mathbf{y}^\pi| + \gamma \mathbb{E}_{\mathbf{x}' \sim P_\mathbf{x}^\pi, \mathbf{y}' \sim P_\mathbf{y}^\pi} d(\mathbf{x}', \mathbf{y}'),$$

*$\mathcal{F}_{MICo}$ has a fixed point $d_{MICo}^\pi$.*

## 3 State Chrono Representation

Although the bisimulation metric [47] and MICo [7] have their strengths, they do not adequately capture future information. This limitation can degrade the effectiveness of state representations in policy learning. To address this issue and incorporate future details, we propose a novel approach, **State Chrono Representation** (SCR). The detailed architecture of SCR is illustrated in Figure 1.

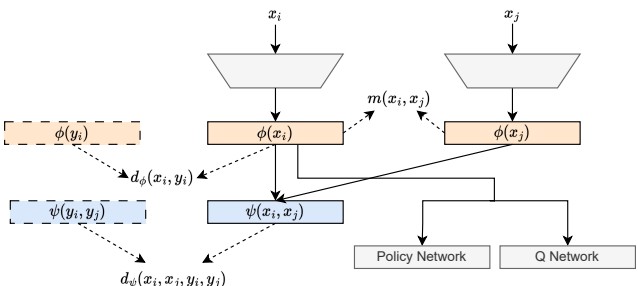

Figure 1: Overall architecture of SCR.

SCR consists of two key components: a **state representation** $\phi(\mathbf{x}) \in \mathbb{R}^n$ for a given state $\mathbf{x}$, and a **chronological embedding** $\psi(\mathbf{x}_i, \mathbf{x}_j) \in \mathbb{R}^n$ that captures the relationship between a current state $\mathbf{x}_i$ and its future state $\mathbf{x}_j$ within the same trajectory. The state representation, $\phi(\mathbf{x})$, is developed using a behavioral metric $d$ that quantifies the reward and dynamic divergence between two states. On the other hand, the chronological embedding, $\psi(\mathbf{x}_i, \mathbf{x}_j)$, fuses these two state representations using deep learning, with a focus on capturing the long-term behavioral correlation between the current state $\mathbf{x}_i$ and the future state $\mathbf{x}_j$. To compute the distance between any two chronological embeddings, we propose a "chronological" behavioral metric, which is further improved through a Bellman operator-like MSE loss [36]. Moreover, $\phi(\mathbf{x})$ incorporates a novel **temporal measurement** $m$ to evaluate the transition from the current state to a future state, effectively capturing sequential reward data from the optimal behavior. Due to the complexity of the regression target, this temporal measurement operates within the defined lower and upper constraints, providing a stable prediction target for both the measurement and state representation during the learning process.

## 3.1 Metric Learning for State Representation

The state representation encoder $\phi$ is trained by approximating a behavioral metric, such as the MICo distance. In our model, we utilize a MICo-based metric transformation operator. Instead of using sampling-based prediction in MICo, we employ latent dynamics-based modeling to assess the divergence between two subsequent state distributions. This approach draws inspiration from the methodology in SimSR [45]. The metric update operator for latent dynamics, denoted by $\mathcal{F}$, is defined as follows.

**Theorem 3.1.** *Let $\hat{d} : \mathbb{R}^n \times \mathbb{R}^n \to \mathbb{R}$ be a metric in the latent state representation space, $d_\phi(\mathbf{x}_i, \mathbf{y}_{i'}) := \hat{d}(\phi(\mathbf{x}_i), \phi(\mathbf{y}_{i'}))$ be a metric in the state domain. The metric update operator $\mathcal{F}$ is defined as,*

$$\mathcal{F}d_\phi(\mathbf{x}_i, \mathbf{y}_{i'}) = |r_{\mathbf{x}_i} - r_{\mathbf{y}_{i'}}| + \gamma \mathbb{E}_{\substack{\phi(\mathbf{x}_{i+1}) \sim \hat{P}(\cdot | \phi(\mathbf{x}_i), a_{\mathbf{x}_i}) \\ \phi(\mathbf{y}_{i'+1}) \sim \hat{P}(\cdot | \phi(\mathbf{y}_{i'}), a_{\mathbf{y}_{i'}}) \\ a_{\mathbf{x}_i}, a_{\mathbf{y}_{i'}} \sim \pi}} \hat{d}(\phi(\mathbf{x}_{i+1}), \phi(\mathbf{y}_{i'+1})), \tag{1}$$

*where $a_{\mathbf{x}_i}$ and $a_{\mathbf{y}_{i'}}$ are the actions in states $\mathbf{x}_i$ and $\mathbf{y}_{i'}$, respectively, and $\hat{P}$ is the learned latent dynamics model. $\mathcal{F}$ has a fixed point $d_\phi^\pi$.*

*Proof.* Refer to Appendix A.3.1 for the detailed proof. □

To learn the approximation for $d_\phi^\pi$ in the embedding space, one needs to specify the form of distance $\hat{d}$ for latent vectors. Castro et al. [7] showed that a behavioral metric with a sample-based next state distribution divergence is a diffuse metric, as the divergence of the next state distribution corresponds to the Łukaszyk-Karmowski distance [49] that measures the expected distance between two samples drawn from two distributions, respectively (detailed definition is in Appendix A.4.1).

**Definition 3.2** (Diffuse metric [7]). A function $d : \mathcal{X} \times \mathcal{X} \to \mathbb{R}$ based on a vector space $\mathcal{X}$ is a diffuse metric if the following axioms hold: 1) $d(\mathbf{a}, \mathbf{b}) \geq 0$ for any $\mathbf{a}, \mathbf{b} \in \mathcal{X}$; 2) $d(\mathbf{a}, \mathbf{b}) = d(\mathbf{b}, \mathbf{a})$ for any $\mathbf{a}, \mathbf{b} \in \mathcal{X}$; 3) $d(\mathbf{a}, \mathbf{b}) + d(\mathbf{b}, \mathbf{c}) \geq d(\mathbf{a}, \mathbf{c})$ for any $\mathbf{a}, \mathbf{b}, \mathbf{c} \in \mathcal{X}$.

MICo provides an approximation of the behavioral metric using an angular distance: $\hat{d}_{MICo}(\mathbf{a}, \mathbf{b}) = \frac{\|\mathbf{a}\|_2^2 + \|\mathbf{b}\|_2^2}{2} + \beta\theta(\mathbf{a}, \mathbf{b})$, where $\mathbf{a}, \mathbf{b} \in \mathbb{R}^n$, $\theta(\mathbf{a}, \mathbf{b})$ represents the angle between vectors $\mathbf{a}$ and $\mathbf{b}$, and $\beta = 0.1$ is a hyperparameter. This distance calculation includes a non-zero self-distance, which makes it compatible with expressing the Łukaszyk-Karmowski distance [49]. However, the angle function $\theta(\mathbf{a}, \mathbf{b})$ only considers the angle between $\mathbf{a}$ and $\mathbf{b}$, which requires computations involving cosine similarity and the arccos function. This can sometimes lead to numerical discrepancies. DBC [47] recommends using the $L_1$ norm with zero self-distance, which is only suitable for the Wasserstein distance. On the other hand, SimSR [45] utilizes the cosine distance, derived from cosine similarity, but it does not satisfy the triangle inequality and does not have a non-zero self-distance.

To address the challenges mentioned above, we propose a revised distance, $\hat{d}(\mathbf{a}, \mathbf{b})$, in the embedding space. It is characterized as a diffuse metric and formulated as follows:

**Definition 3.3.** We define $\hat{d} : \mathbb{R}^n \times \mathbb{R}^n \to \mathbb{R}$ as a distance function, where $\hat{d}(\mathbf{a}, \mathbf{b}) = \sqrt{\|\mathbf{a}\|_2^2 + \|\mathbf{b}\|_2^2 - \mathbf{a}^\top \mathbf{b}}$, for any $\mathbf{a}, \mathbf{b} \in \mathbb{R}^n$.

**Theorem 3.4.** *$\hat{d}$ is a diffuse metric.*

*Proof.* Refer to Appendix A.3.2 for the detailed proof. □

**Lemma 3.5** (Non-zero self-distance). *The self-distance of $\hat{d}$ is not strict to zero, i.e., $\hat{d}(\mathbf{a}, \mathbf{a}) = \|\mathbf{a}\|_2 \geq 0$. It becomes zero if and only if every element in vector $\mathbf{a}$ is zero.*

Theorem 3.4 validates that $\hat{d}$ is a diffuse metric satisfying triangle inequality, and Lemma 3.5 shows $\hat{d}$ has non-zero self-distance capable of approximating dynamic divergence, which is a Łukaszyk-Karmowski distance. Moreover, the structure of $\hat{d}$ closely resembles the $L_2$ norm, with the only difference being that the factor of $\mathbf{a}^\top \mathbf{b}$ is $-1$ instead of $-2$. This construction, which involves only vector inner product and square root computations without divisions or trigonometric functions, helps avoid numerical computational issues and simplify the implementation.

To learn the representation function $\phi$, a common approach is to minimize MSE loss between the two ends of (1). The loss, which incorporates $\hat{d}$ and is dependent on $\phi$, can be expressed as follows:

$$\mathcal{L}_\phi(\phi) = \mathbb{E}_{\substack{\mathbf{x}_i,\mathbf{y}_{i'},r_{\mathbf{x}_i},r_{\mathbf{y}_{i'}}\sim\mathcal{D} \\ \phi(\mathbf{x}_{i+1}),\phi(\mathbf{y}_{i'+1})\sim\hat{P}}} \left| \hat{d}(\phi(\mathbf{x}_i),\phi(\mathbf{y}_{i'})) - |r_{\mathbf{x}_i} - r_{\mathbf{y}_{i'}}| - \gamma\hat{d}(\phi(\mathbf{x}_{i+1}),\phi(\mathbf{y}_{i'+1})) \right|^2, \quad (2)$$

where $\mathcal{D}$ represents the replay buffer or the sampled rollouts used in the RL learning process.

### 3.1.1 Long-term Temporal Information

The loss $\mathcal{L}_\phi(\phi)$ in Eqn. (2) heavily relies on the temporal-difference update mechanism, utilizing only one-step transition information. However, this approach fails to effectively capture and encode long-term information from trajectories. Incorporating temporal information into the representation while maintaining the structure and adhering to behavioral metrics presents a complex challenge. To address this challenge, we propose two distinct methods: Chronological Embedding (Section 3.2) and Temporal Measurement (Section 3.3). Each technique is designed to capture the temporal essence of a rollout denoted as

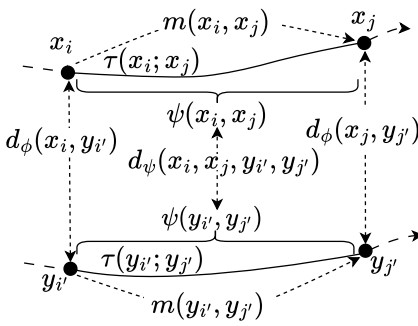

Figure 2: An example with two rollouts.

$\tau(\mathbf{x}_i;\mathbf{x}_j)$, which represents a sequence starting from state $\mathbf{x}_i$ and reaching a future state $\mathbf{x}_j$.

As illustrated in Figure 2, the goal of the chronological embedding is to create a novel paired state embedding, denoted as $\psi(\mathbf{x}_i,\mathbf{x}_j)$, capturing the temporal representation of the rollout $\tau(\mathbf{x}_i;\mathbf{x}_j)$. This is achieved by transforming a pair of state representations $\phi(\mathbf{x}_i)$ and $\phi(\mathbf{x}_j)$ to a vector: $\psi(\mathbf{x}_i,\mathbf{x}_j):=\psi(\phi(\mathbf{x}_i),\phi(\mathbf{x}_j)) \in \mathbb{R}^n$. The objective of learning $\psi$ is to develop a "chronological" behavioral metric $d_\psi$ that quantifies the distance between rollouts $\tau(\mathbf{x}_i;\mathbf{x}_j)$ and $\tau(\mathbf{y}_{i'};\mathbf{y}_{j'})$. On the other hand, temporal measurement aims to compute a specific "distance" $m$ between states $\mathbf{x}_i$ and $\mathbf{x}_j$, which provides insights into the cumulative rewards accumulated during the rollout $\tau(\mathbf{x}_i;\mathbf{x}_j)$.

Note that learning the state representation solely based on either $d_\psi$ or $m$ poses challenges. While $d_\psi$ improves the representation expressiveness, capturing both rewards and dynamics may be less efficient in the learning process. On the other hand, learning $m$ is efficient but may not adequately capture the dynamics. Therefore, we propose a unified approach that leverages the strengths of both $d_\psi$ and $m$ to overcome the challenges and improve the overall state representation. Empirical results in Section 4.3 support the effectiveness of our proposal.

## 3.2 Chronological Embedding

The **chronological embedding**, denoted by $\psi(\mathbf{x}_i,\mathbf{x}_j) \in \mathbb{R}^n$, is designed to capture the relationship between a given state $\mathbf{x}_i$ and any of its future states $\mathbf{x}_j$ over the same trajectory. To capture the comprehensive behavioral knowledge, we introduce a distance function $d_\psi : (\mathcal{S} \times \mathcal{S}) \times (\mathcal{S} \times \mathcal{S}) \to \mathbb{R}$, which is intended to reflect the behavioral metric and enable the encoder $\psi$ to incorporate behavioral information. Building upon the MICo distance in Theorem 2.2, we specify the metric update operator $\mathcal{F}_{Chrono}$ for $d_\psi$ in the following theorem. Proof is detailed in Appendix A.3.3.

**Theorem 3.6.** *Let $\mathbb{M}_\psi$ be the space of $d_\psi$. The metric update operator $\mathcal{F}_{Chrono} : \mathbb{M}_\psi \to \mathbb{M}_\psi$ is defined as follows,*

$$\mathcal{F}_{Chrono}d_\psi(\mathbf{x}_i,\mathbf{x}_j,\mathbf{y}_{i'},\mathbf{y}_{j'}) = |r_{\mathbf{x}_i} - r_{\mathbf{y}_{i'}}| + \gamma\mathbb{E}_{\mathbf{x}_{i+1}\sim P_\mathbf{x}^\pi,\mathbf{y}_{i'+1}\sim P_\mathbf{y}^\pi}d_\psi(\mathbf{x}_{i+1},\mathbf{x}_j,\mathbf{y}_{i'+1},\mathbf{y}_{j'}), \quad (3)$$

*where time step $i < j$ and $i' < j'$. $\mathcal{F}_{Chrono}$ has a fixed point $d_\psi^\pi$.*

Here $d_\psi^\pi$ represents the "chronological" behavioral metric. Our objective is to closely approximate $d_\psi^\pi$, which measures the distance between two sets of states $(\mathbf{x}_i,\mathbf{x}_j)$ and $(\mathbf{y}_{i'},\mathbf{y}_{j'})$, taking into account the difference in immediate rewards and dynamics divergence. To co-learn the encoder $\psi$ with $d_\psi^\pi$, we represent $d_\psi^\pi$ as $d_\psi^\pi(\mathbf{x}_i,\mathbf{x}_j,\mathbf{y}_{i'},\mathbf{y}_{j'}) := \hat{d}(\psi((\mathbf{x}_i,\mathbf{x}_j)),\psi(\mathbf{y}_{i'},\mathbf{y}_{j'}))$, where $\hat{d}$ is defined in Definition 3.3. Similar to Eqn. (1), we construct $d_\psi^\pi$ to compute the distance in the embedding space.

$$d_\psi^\pi(\mathbf{x}_i,\mathbf{x}_j,\mathbf{y}_{i'},\mathbf{y}_{j'}) = |r_{\mathbf{x}_i} - r_{\mathbf{y}_{i'}}| + \gamma\mathbb{E}_{\mathbf{x}_{i+1}\sim P_\mathbf{x}^\pi,\mathbf{y}_{i'+1}\sim P_\mathbf{y}^\pi}\hat{d}(\psi(\mathbf{x}_{i+1},\mathbf{x}_j),\psi(\mathbf{y}_{i'+1},\mathbf{y}_{j'})). \quad (4)$$

To ensure computational efficiency, the parameters of the encoders $\phi$ and $\psi$ are shared. The encoder $\psi$ extracts outputs from $\phi$, and the distance measure is appropriately adapted as $d_\psi^\pi(\mathbf{x}_i, \mathbf{x}_j, \mathbf{y}_{i'}, \mathbf{y}_{j'}) := \hat{d}(\psi(\phi(\mathbf{x}_i), \phi(\mathbf{x}_j)), \psi(\phi(\mathbf{y}_{i'}), \phi(\mathbf{y}_{j'})))$. The objective of learning the chronological embedding is to minimize the MSE loss between both sides of Eqn. (4) w.r.t. $\phi$ and $\psi$,

$$\mathcal{L}_\psi(\psi, \phi) = \mathbb{E}_{\substack{\mathbf{x}_i, \mathbf{x}_j, \mathbf{y}_{i'}, \mathbf{y}_{j'}, r_{\mathbf{x}_i}, \\ r_{\mathbf{y}_{i'}}, \mathbf{x}_{i+1}, \mathbf{y}_{i'+1} \sim \mathcal{D}}} \left| \hat{d}(\psi(\phi(\mathbf{x}_i), \phi(\mathbf{x}_j)), \psi(\phi(\mathbf{y}_{i'}), \phi(\mathbf{y}_{j'}))) - |r_{\mathbf{x}_i} - r_{\mathbf{y}_{i'}}| \right.$$
$$\left. -\gamma \hat{d}(\psi(\phi(\mathbf{x}_{i+1}), \phi(\mathbf{x}_j)), \psi(\phi(\mathbf{y}_{i'+1}), \phi(\mathbf{y}_{j'}))) \right|^2. \qquad (5)$$

Minimizing Eqn. (5) is to encourage the embeddings of similar state sequences to become closer in the embedded space, thereby strengthening the classification of similar behaviors.

### 3.3 Temporal Measurement

To enhance the ability of SCR to gain future insights, we introduce the concept of **temporal measurement**, which quantifies the disparities between a current state $\mathbf{x}_i$ and a future state $\mathbf{x}_j$. This measurement, denoted by $m(\mathbf{x}_i, \mathbf{x}_j)$, aims to measure the differences in state values or the cumulative rewards obtained between the current and future states. We construct an approximate measurement based on the state representation $\phi$, denoted by $\hat{m}_\phi(\mathbf{x}_i, \mathbf{x}_j) := \hat{m}(\phi(\mathbf{x}_i), \phi(\mathbf{x}_j))$. Here, $\hat{m} : \mathbb{R}^n \times \mathbb{R}^n \to \mathbb{R}$ is a non-parametric asymmetric metric function (details are presented at the end of this section). This approach structures the representation space of $\phi(\mathbf{x})$ around the "distance" $m$, ensuring that $\phi(\mathbf{x})$ contains sufficient information for future state planning.

We propose using $m$ to represent the expected discounted accumulative rewards obtained by an optimal policy $\pi^*$ from state $\mathbf{x}_i$ to state $\mathbf{x}_j$: $m(\mathbf{x}_i, \mathbf{x}_j) = \mathbb{E}_{\pi^*}\left[\sum_{t=0}^{j-i} \gamma^t r_{\mathbf{s}_t} \middle| \mathbf{s}_0 = \mathbf{x}_i, \mathbf{s}_{j-i} = \mathbf{x}_j\right]$. However, obtaining the exact value of $m(\mathbf{x}_i, \mathbf{x}_j)$ is challenging because the optimal policy $\pi^*$ is unknown and is, in fact, the primary goal of the RL task. Instead of directly approximating $m(\mathbf{x}_i, \mathbf{x}_j)$, we learn the approximation $\hat{m}(\mathbf{x}_i, \mathbf{x}_j)$ in an alternative manner that ensures it falls within a feasible range covering the true $m(\mathbf{x}_i, \mathbf{x}_j)$.

To construct this range, we introduce two constraints. The first constraint, serving as a **lower** boundary, states that the expected discounted cumulative reward obtained by any policy $\pi$, whether optimal or not, cannot exceed $m$:

$$\mathbb{E}_\pi\left[\sum_{t=0}^{j-i} \gamma^t r_{\mathbf{s}_t} \middle| \mathbf{s}_0 = \mathbf{x}_i, \mathbf{s}_{j-i} = \mathbf{x}_j\right] \leq m(\mathbf{x}_i, \mathbf{x}_j). \qquad (6)$$

This constraint is based on a fact that any sub-optimal policy is inferior to the optimal policy. Based on this constraint, we propose the following objective to learn the approximation $\hat{m}_\phi$:

$$\mathcal{L}_{low}(\phi) = \mathbb{E}_{\tau(\mathbf{x}_i; \mathbf{x}_j) \sim \pi} \left| \text{ReLU}\left(\sum_{t=0}^{j-i} \gamma^t r_{\mathbf{x}_t} - \hat{m}(\phi(\mathbf{x}_i), \phi(\mathbf{x}_j))\right) \right|^2, \qquad (7)$$

where $\text{ReLU}(x) = \max(0, x)$. This objective is non-zero when the constraint in Eqn. (6) is not satisfied. Hence, it increases the value of $m(\phi(\mathbf{x}_i), \phi(\mathbf{x}_j))$ until it exceeds the sampled reward sum.

The second constraint serving as the upper boundary is proposed based on Fig. 2. To ensure that the absolute value $|m(\mathbf{x}_i, \mathbf{x}_j)|$ remains within certain limits, we propose the following inequality:

$$|\hat{m}(\mathbf{x}_i, \mathbf{x}_j)| \leq d(\mathbf{x}_i, \mathbf{y}_{i'}) + |\hat{m}(\mathbf{y}_{i'}, \mathbf{y}_{j'})| + d(\mathbf{x}_j, \mathbf{y}_{j'}), \qquad (8)$$

where $d$ is the behavioral metric introduced in Section 3.1. The right-hand side of the inequality represents the longer path from $\mathbf{x}_i$ to $\mathbf{x}_j$. This inequality demonstrates that the absolute temporal measurement $|m(\mathbf{x}_i, \mathbf{x}_j)|$ should not exceed the sum of the behavioral metrics at the initial states ($\mathbf{x}_i$ and $\mathbf{y}_{i'}$), i.e., $d(\mathbf{x}_i, \mathbf{y}_{i'})$, the final states pair ($\mathbf{x}_j$ and $\mathbf{y}_{j'}$), i.e. $d(\mathbf{x}_j, \mathbf{y}_{j'})$, and the measurement $|m(\mathbf{y}_i, \mathbf{y}_j)|$. This constraint leads to the following objective for training $\hat{m}_\phi$:

$$\mathcal{L}_{up}(\phi) = \left| \text{ReLU}\left(|\hat{m}(\phi(\mathbf{x}_i), \phi(\mathbf{x}_j))| - \text{sg}\left(\hat{d}(\phi(\mathbf{x}_i), \phi(\mathbf{y}_{i'})) + \hat{d}(\phi(\mathbf{x}_j), \phi(\mathbf{y}_{j'})) + \hat{m}(\phi(\mathbf{y}_{i'}), \phi(\mathbf{y}_{j'}))\right)\right) \right|^2,$$

where sg means "stop gradient". This objective aims to decrease the value of $\hat{m}_\phi$ when the constraint in Eqn. (8) is not satisfied. By simultaneously optimizing both constraints in a unified manner,

we guarantee that the approximated temporal measurement, $m$, is confined within a specific range, defined by the lower and upper constraints. The overall objective for $\hat{m}$ is formulated as follows:

$$\mathcal{L}_{\hat{m}}(\phi) = \mathcal{L}_{low}(\phi) + \mathcal{L}_{up}(\phi). \tag{9}$$

**Asymmetric Metric Function for $\hat{m}$.** The measurement $\hat{m}(\mathbf{x}_i, \mathbf{x}_j)$, designed to measure the distance based on the rewards, is intended to be **asymmetric** with respect to $\mathbf{x}_i$ and $\mathbf{x}_j$. This is because we assume that state $\mathbf{x}_i$ comes before $\mathbf{x}_j$, resulting in a distinct relationship compared to the progression from $\mathbf{x}_j$ to $\mathbf{x}_i$. Recent research has focused on studying quasimetrics in deep learning [28, 39, 38] and developed various methodologies to compute asymmetric distances. In our method, we choose to utilize Interval Quasimetric Embedding (IQE) [38] to implement $\hat{m}$ (details are in Appendix A.5.1).

## 3.4 Overall Objective

As shown in Figure 1, the encoders are designed to predict state representations $\phi(\mathbf{x})$ for individual states and chrono embedding $\psi(\mathbf{x}_i, \mathbf{x}_j)$ to capture the relationship between states $\mathbf{x}_i$ and $\mathbf{x}_j$. The measurement $\hat{m}$ is then computed based on $\phi(\mathbf{x}_i)$ and $\phi(\mathbf{x}_j)$ to incorporate the accumulated rewards between these states. The components $\psi$ and $\hat{m}$ work together to enhance the state representations $\phi$ and capture temporal information, thereby improving their predictive capabilities for future insights. The necessity of $\psi$ and $\hat{m}$ is demonstrated in the ablation study in Section 4.3. Therefore, a comprehensive objective to minimize is formulated in a unified manner as follows,

$$\mathcal{L}(\phi, \psi) = \mathcal{L}_{\phi}(\phi) + \mathcal{L}_{\psi}(\psi, \phi) + \mathcal{L}_{\hat{m}}(\phi). \tag{10}$$

Our method, SCR, can be seamlessly integrated with a wide range of deep RL algorithms, which can effectively utilize the representation $\phi(\mathbf{x})$ as a crucial input component. In our implementation, we specifically employ Soft Actor-Critic (SAC) [13] as our foundation RL algorithm. The state representation serves as the input state for both the policy network and Q-value network in SAC. Other implementation details can be found in Appendix A.5.

# 4 Experiments

## 4.1 Configurations

**DeepMind Control Suite.** The primary objective of our proposed SCR is to develop a robust and generalizable representation for deep RL when dealing with high-dimensional observations. To evaluate its effectiveness, we conduct experiments using the DeepMind Control Suite (DM_Control) environment, which involves rendered pixel observations [37] and a distraction setting called Distracting Control Suite [34]. This environment utilizes the MuJoCo engine, offering pixel observations for continuous control tasks. The DM_Control environment provides diverse testing scenarios, including background and camera pose distractions, simulating real-world complexities of camera inputs. (1) *Default setting* evaluates the effectiveness of each RL approach. Frames are rendered at a resolution of $84 \times 84$ RGB pixels as shown in Figure 3(a). We stack three frames as states for the RL agents. (2) *Distraction setting* evaluates both the generalization and effectiveness of each RL approach. The distraction [34] consists of 3 components, as shown in Figure 3(b): 1) **background video** distraction, where the clean and simple background is replaced with a natural video; 2) **object color** distraction, which involves slight changes in the color of the robots; and 3) **camera pose** distraction, where the camera's position and angle are randomized during rendering from the simulator. We observe that tasks become significantly more challenging when the camera pose distraction is applied.

**Baselines.** We compare our method against several prominent algorithms in the field, including: 1) SAC [13], a baseline deep RL method for continuous control; 2) DrQ [42], a data augmentation

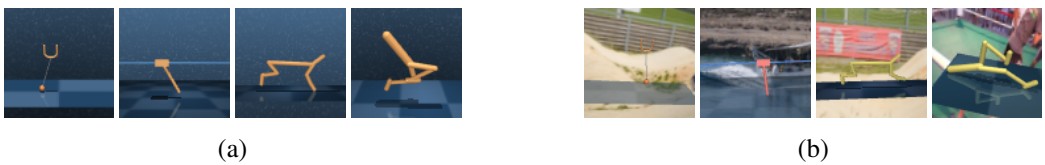

(a)                                       (b)

Figure 3: Examples of DM_Control with (a) default setting and (b) distraction setting.

| | BiC-Catch | C-SwingUp | C-SwingUpSparse | Ch-Run | F-Spin | H-Stand | R-Easy | W-Walk |
|---|---|---|---|---|---|---|---|---|
| SAC | 447.5±41.4 | 631.3±103.6 | 30.1±12.7 | 354.4±15.4 | 518.6±29.3 | 739.8±15.6 | 321.1±95.4 | 363.7±154.4 |
| DrQ | **962.7**±19.6 | 840.3±8.2 | **760.9**±44.6 | 487.6±13.9 | 962.0±11.3 | **861.2**±20.9 | **970.2**±12.7 | **927.3**±14.5 |
| DBC | 104.2±33.3 | 281.3±27.6 | 65.9±80.3 | 386±16.6 | 730.3±136.9 | 43.5±55.6 | 179.8±27.4 | 330.3±41.4 |
| MICo | 215.4±12.4 | 803.2±12.0 | 0.0±0.0 | 4.9±1.8 | 2.0±0.1 | 800.8±21.1 | 186.1±19.4 | 29.7±3.3 |
| SimSR | 938.8±14.9 | **854.8**±11.5 | 217.9±307.5 | 255.3±327.5 | **962.4**±19.0 | 6.2±1.7 | 76.6±25.1 | **929.9**±21.1 |
| SCR | 944.2±11.7 | **849.4**±2.1 | **768.4**±15.1 | **778.4**±29.7 | **964.9**±25.5 | 851.3±45.6 | 946.8±26.0 | **919.0**±20.0 |

Table 1: Results (mean±std) on DM_Control with the default setting at 500K environment steps.

| | BiC-Catch | C-SwingUp | C-SwingUpSparse | Ch-Run | F-Spin | H-Stand | R-Easy | W-Walk |
|---|---|---|---|---|---|---|---|---|
| SAC | 82.6±20.2 | 218.4±4.9 | 0.7±0.7 | 177.4±8.4 | 22.5±27.7 | 19.6±26.2 | 149.4±77.5 | 167.2±8 |
| DrQ | 124.0±99.6 | 230.0±36.3 | 11.2±6.4 | 103.0±84.9 | 579.8±282.8 | 16.8±11.8 | 70.5±40.1 | 33.6±6.3 |
| DBC | 48.8±24.4 | 127.7±19.2 | 0.0±0.0 | 9.2±1.9 | 7.7±10.7 | 5.6±2.5 | 149.6±42.8 | 30.9±4.8 |
| MICo | 104.2±10 | 200.2±6.6 | 0.0±0.1 | 7.4±3.2 | 86.9±76.1 | 11.8±10.5 | 132.3±37.8 | 27.5±7.7 |
| SimSR | 106.4±13.5 | 148.4±17.3 | 0.0±0.0 | 28.2±23.8 | 0.4±0.1 | 6.6±1.2 | 78.4±17 | 28.6±3.1 |
| SCR | **221.3**±55.4 | **565.1**±59.9 | **185.7**±93.0 | **331.7**±1.4 | **738.3**±24.5 | **400.8**±19.2 | **666.1**±14.5 | **555.1**±31.2 |

Table 2: Results (mean±std) on DM_Control with distraction setting at 500K environment step. Distraction includes background, robot body color, and camera pose.

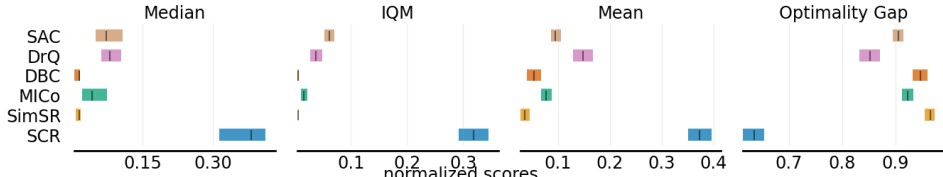

Figure 4: Aggregate metrics on distract setting.

method using random crop; 3) DBC [47], representation learning with the bisimulation metric; 4) MICo [7], representation learning with MICo distance; and 5) SimSR [45], representation learning with the behavioral metric approximated by the cosine distance.

## 4.2 Main Results

**Default Setting.** To verify the sample efficiency of our method, we compare it with others on eight tasks in DM_Control and report the experimental results in Table 1. We trained each method on each task for 500K environment steps. All results are averaged over 10 runs. We observe that SCR achieves comparable results to the augmentation method DrQ and the state-of-the-art behavioral metric approach SimSR. It is worth noting that for a DM_Control task, the maximum achievable scores are 1000, and a policy that collects scores around 900 is considered nearly optimal. These findings highlight the effectiveness of our SCR in mastering standard RL control tasks.

**Distraction Setting.** To further evaluate the generalization ability of our method, we perform comparison experiments on DM_Control with Distraction using the same training configuration as in the default setting. We evaluate 10 different seeds, and conduct 100 episodes per run. The scores from each run are summarized in Table 2. The std is computed across the mean scores of these 10 runs. The results show that our method outperforms the others in all eight tasks, notably in the sparse reward task *cartpole-swing_up_sparse*, where other methods achieve nearly zero scores. The camera-pose distraction poses a significant challenge for metric-based methods like DBC, MICo, and SimSR, as it leads to substantial distortion of the robot's shape and position in the image state. DrQ outperforms other behavioral metric methods, possibly due to its random cropping technique, which facilitates better alignment of the robot's position. The aggregate metrics [2] are shown in Figure 4, where scores are normalized by dividing by 1000. Figure 5 presents the training curves, while additional curves can be found in Appendix B.3. These results demonstrate the superior performance of our method in handling distractions and highlight our strong generalization capabilities.

## 4.3 Ablation Study

**Impact of Different Components.** To evaluate the impact of each component in SCR, we perform an ablation study where certain components are selectively removed or substituted. Figure 6 shows the training curves on *cheetah-run* and *walker-walk* in the distraction setting. SCR denotes the full

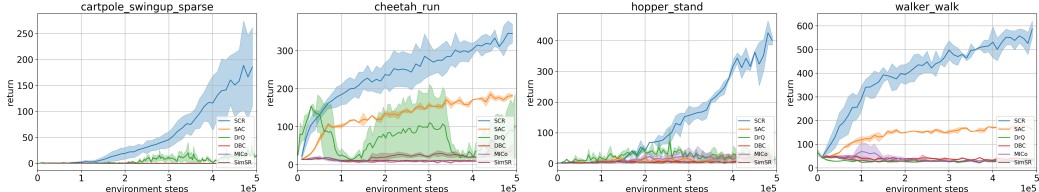

Figure 5: Training curves of SCR and baseline methods in the distraction setting of DM_Control. Mean scores on 10 runs with std (shadow shape). Training curves of all tasks are shown in Appendix B.3.

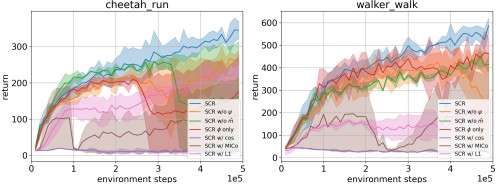

Figure 6: Ablation study on *cheetah-run* (left) and *walker-walk* (right) in the distraction setting. Mean scores on 10 runs with std (shadow shape).

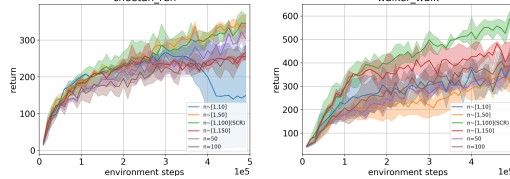

Figure 7: Training curves with varying sampling steps. Left: *cheetah-run*. Right: *walker-walk*. Mean scores on 10 runs with std (shadow shape).

model. SCR **w/o** $\psi$ removes the chronological embedding $\psi$. SCR **w/o** $\hat{m}$ refers to the exclusion of the approximation $\hat{m}$. SCR **w/** $\phi$ **only** removes losses $\mathcal{L}_\psi(\phi, \psi)$ and $\mathcal{L}_{\hat{m}}(\phi)$ but keep $\mathcal{L}_\phi(\phi)$. SCR **w/ cos** replaces the distance function $\hat{d}$ for computing metrics on the representation space with cosine distance, akin to SimSR does. SCR **w/ MICo** replaces $\hat{d}$ with MICo's angular distance. SCR **w/ L1** replaces $\hat{d}$ with the $L_1$ distance as adopted by DBC. The results demonstrate the superior performance of the full model and the importance of $\psi$ and $\hat{m}$. The absence of these components can lead to worse performance and unstable training. The results also demonstrate that the proposed $\hat{d}$ surpasses existing distance functions.

**Impact of Sampling Steps.** In our previous experiments, we uniformly sample the number of steps between $i$ and $j$ in the range $[1, 100]$. In this experiment, we investigate the impact of this hyper-parameter on SCR. We conduct experiments on the *cheetah-run* and *walker-walk* tasks in the distraction setting with various step sampling ranges: $[1, 10]$, $[1, 50]$, $[1, 100]$, and $[1, 150]$. We also make comparisons with fixed step counts: 50 and 100. The results presented in Figure 7 show that sampling step counts in the range $[1, 100]$ yields the optimal results and remains stable across tasks. Therefore, we set this hyper-parameter to uniformly sample in the range $[1, 100]$ for all experiments.

### 4.4 Experiments on Meta-World

We present experimental investigations conducted in Meta-World [44], a comprehensive simulated benchmark that includes distinct robotic manipulation tasks. Our focus is on six specific tasks: *basketball-v2*, *coffee-button-v2*, *door-open-v2*, *drawer-open-v2*, *pick-place-v2*, and *window-open-v2*, chosen for their diverse objects in environments. The observations are rendered as $84 \times 84$ RGB pixels with a frame stack of 3, following the DM_Control format. Table 3 displays the average success rates over all tasks and five seeds, while Figure 8 shows the training curves for a subset of these tasks. Additional training curves for all tasks can be found in Appendix B.5. Our proposed SCR consistently achieves optimal performance in all tasks, while other baseline methods, except for DrQ, fail to converge to such high levels of performance. Even though DrQ demonstrates proficiency in achieving optimal performance, it shows less sampling efficiency than SCR, highlighting the effectiveness of SCR in the applied setting.

## 5 Conclusion

Our proposed State Chrono Representation learning framework (SCR) effectively captures information from complex, high-dimensional, and noisy inputs in deep RL. SCR improves upon previous metric-based approaches by incorporating a long-term perspective and quantifying state distances across

| | SAC | DrQ | DBC | MICo | SimSR | SCR |
|---|---|---|---|---|---|---|
| Meta-World | $0.495 \pm 0.475$ | $0.886 \pm 0.125$ | $0.479 \pm 0.453$ | $0.495 \pm 0.482$ | $0.258 \pm 0.365$ | $\mathbf{0.969} \pm 0.032$ |

Table 3: Average success rates for six tasks in Meta-World.

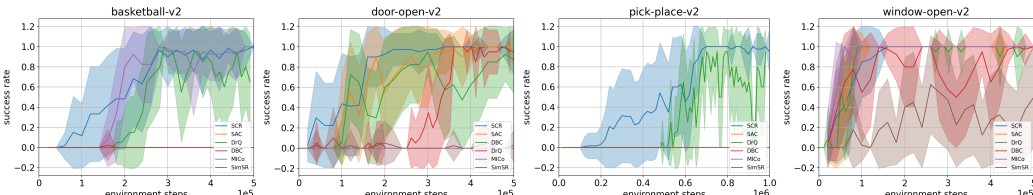

Figure 8: Training curves in Meta-World. Mean success rates on 5 runs with std (shadow shape). Training curves of all tasks are shown in Appendix B.5.

temporal trajectories. Our extensive experiments demonstrate its effectiveness compared with several metric-based baselines in complex environments with distractions, making a promising avenue for future research on generalization in representation learning.

**Limitations.** This work does not address truly Partially Observable MDPs (POMDPs), which are common in real-world applications. In future work, we plan to integrate SCR with POMDP methods to solve real-world problems, or to design unified solutions that better accommodate these complexities.

## 6   Acknowledgement

We thank the anonymous reviewers for the constructive feedback. This study is supported under RIE2020 Industry Alignment Fund – Industry Collaboration Projects (IAF-ICP) Funding Initiative, as well as cash and in-kind contribution from the industry partner(s). Sinno J. Pan thanks the support of the Hong Kong Jockey Club Charities Trust to the JC STEM Lab of Integration of Machine Learning and Symbolic Reasoning.

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

# A Appendix

## A.1 Broader Impact

This paper is a research on improving the representation of deep reinforcement learning. We think that there are no potential societal consequences that need to be highlighted.

## A.2 Additional Related Work

Recent studies have extensively explored representation learning in reinforcement learning (RL). Some previous works [18, 24, 43] employ autoencoders to encode image states into low-dimensional latent embeddings, resulting in improved visual perception and accelerated policy learning. Other approaches [42, 22, 23, 41, 35] utilize data augmentation techniques such as random crop or noise injection, combined with contrastive loss, to learn more generalizable state representations. Another set of approaches [25, 33, 15] focus on learning representation by predicting auxiliary tasks to extract more information from the environment. Successor representations [19, 30] tackles the state occupancy distribution to learn a more informative state representation, showing generalization capabilities between tasks.

In recent research, metric learning methods have been employed in RL to measure distances between state representations. Some approaches aim to approximate bisimulation metrics [47, 20] while others focus on learning sample-based distances [45, 7]. Bisimulation metrics have also been utilized in goal-based RL to enhance state representation [17]. A more recent work introduces quasimetrics learning as a novel RL objective for cost MDPs [40] but it is not for general MDPs.

Self-predictive representation (SPR) [32] learn representations regarding a future state but it focuses on predicting future states, enforcing the representation of predicted future states to be close to the true future states. SPR's representation focuses on dynamics learning, without directly considering the reward or value. In contrast, SCR focuses on predicting a metric/distance between current state and future state and the metrics are related to rewards or values regarding policy learning.

Successor Representations (SRs) [1, 3, 30] are a class of methods designed to learn representations that facilitate generalization. SRs achieve this by focusing on the occupancy of states, enabling generalization across various tasks and reward functions. In contrast, SCR measures the distance between states specifically to handle observable distractions.

## A.3 Proofs

### A.3.1 Proof of Theorem 3.1

**Theorem 3.** *Let $\hat{d} : \mathbb{R}^n \times \mathbb{R}^n \to \mathbb{R}$ be a metric in the latent state representation space, $d_\phi(\mathbf{x}_i, \mathbf{y}_{i'}) := \hat{d}(\phi(\mathbf{x}_i), \phi(\mathbf{y}_{i'}))$ be a metric in the state domain. The metric update operator $\mathcal{F}$ is defined as,*

$$\mathcal{F}d_\phi(\mathbf{x}_i, \mathbf{y}_{i'}) = |r_{\mathbf{x}_i} - r_{\mathbf{y}_{i'}}| + \gamma \mathbb{E}_{\substack{\phi(\mathbf{x}_{i+1}) \sim \hat{P}(\cdot|\phi(\mathbf{x}_i), a_{\mathbf{x}_i}) \\ \phi(\mathbf{y}_{i'+1}) \sim \hat{P}(\cdot|\phi(\mathbf{y}_{i'}), a_{\mathbf{y}_{i'}})}} \hat{d}(\phi(\mathbf{x}_{i+1}), \phi(\mathbf{y}_{i'+1})), \qquad (11)$$

*where $\hat{\mathbb{M}}$ is the space of $d$, with $a_{\mathbf{x}_i}$ and $a_{\mathbf{y}_{i'}}$ being the actions at states $\mathbf{x}_i$ and $\mathbf{y}_{i'}$, respectively, and $\hat{P}$ is the learned latent dynamics model. $\mathcal{F}$ has a fixed point $d_\phi^\pi$.*

*Proof.* We follow the proof techniques from [7] and [45]. By substituting $\hat{d}(\phi(\mathbf{x}_{i+1}), \phi(\mathbf{y}_{i'+1}))$ with $d_\phi(\mathbf{x}_{i+1}, \mathbf{y}_{i'+1})$, we have

$$\mathcal{F}d_\phi(\mathbf{x}_i, \mathbf{y}_{i'}) = |r_{\mathbf{x}_i} - r_{\mathbf{y}_{i'}}| + \gamma \mathbb{E}_{\substack{\phi(\mathbf{x}_{i+1}) \sim \hat{P}(\cdot|\phi(\mathbf{x}_i), a_{\mathbf{x}_i}) \\ \phi(\mathbf{y}_{i'+1}) \sim \hat{P}(\cdot|\phi(\mathbf{y}_{i'}), a_{\mathbf{y}_{i'}})}} d_\phi(\mathbf{x}_{i+1}, \mathbf{y}_{i'+1}). \qquad (12)$$

The operator $\mathcal{F}d_\phi$ is a contraction mapping with respect to the $L_\infty$ norm because,

$$
\begin{aligned}
|\mathcal{F}d_\phi(\mathbf{x}, \mathbf{y}) - \mathcal{F}d_{\phi'}(\mathbf{x}, \mathbf{y})| &= \left| \gamma \mathbb{E}_{\substack{\phi(\mathbf{x}_{i+1}) \sim \hat{P}(\cdot|\phi(\mathbf{x}_i), a_{\mathbf{x}_i}) \\ \phi(\mathbf{y}_{i'+1}) \sim \hat{P}(\cdot|\phi(\mathbf{y}_{i'}), a_{\mathbf{y}_{i'}})}} (d_\phi - d_{\phi'})(\mathbf{x}_{i+1}, \mathbf{y}_{i'+1}) \right| \\
&\leq \gamma \|(d_\phi - d_{\phi'})(\mathbf{x}_{i+1}, \mathbf{y}_{i'+1})\|_\infty.
\end{aligned} \qquad (13)
$$

By Banach's fixed point theorem, operator $\mathcal{F}$ has a fixed point $d_\phi^\pi$. □

### A.3.2 Proof of Theorem 3.4

To prove the distance $\hat{d}$ is a diffuse metric, we need to prove that $\hat{d}$ satisfies all of three axioms in the Definition 3.2 (definition of diffuse metric).

**Lemma A.1.** $\|\mathbf{a}\|_2^2 + \|\mathbf{b}\|_2^2 - \mathbf{a}^\top \mathbf{b} \geq 0$ *for any* $\mathbf{a}, \mathbf{b} \in \mathbb{R}$.

*Proof.* Because $\mathbf{a}^\top \mathbf{b} \leq \|\mathbf{a}\|\|\mathbf{b}\|$, we have,

$$
\begin{aligned}
&\|\mathbf{a}\|^2 + \|\mathbf{b}\|^2 - \mathbf{a}^\top \mathbf{b} \\
\geq & \|\mathbf{a}\|^2 + \|\mathbf{b}\|^2 - \|\mathbf{a}\|\|\mathbf{b}\| \\
\geq & \|\mathbf{a}\|^2 + \|\mathbf{b}\|^2 - 2\|\mathbf{a}\|\|\mathbf{b}\| \\
= & (\|\mathbf{a}\| - \|\mathbf{b}\|)^2 \\
\geq & 0.
\end{aligned}
\tag{14}
$$

$\square$

This lemma indicates that the term under the square root of $\hat{d}$ is always non-negative. $\hat{d}$ is able to measure any two vectors $\mathbf{a}, \mathbf{b} \in \mathbb{R}^n$.

**Lemma A.2** (Non-negative). $\hat{d}(\mathbf{a}, \mathbf{b}) \geq 0$ *for any* $\mathbf{a}, \mathbf{b} \in \mathbb{R}$.

*Proof.* By definition, the square root is non-negative. $\square$

**Lemma A.3** (Symmetric). $\hat{d}(\mathbf{a}, \mathbf{b}) = \hat{d}(\mathbf{b}, \mathbf{a})$

*Proof.* $\hat{d}(\mathbf{a}, \mathbf{b}) = \sqrt{\|\mathbf{a}\|_2^2 + \|\mathbf{b}\|_2^2 - \mathbf{a}^\top \mathbf{b}} = \sqrt{\|\mathbf{b}\|_2^2 + \|\mathbf{a}\|_2^2 - \mathbf{b}^\top \mathbf{a}} = \hat{d}(\mathbf{b}, \mathbf{a})$ $\square$

**Lemma A.4** (Triangle inequality). $\hat{d}(\mathbf{a}, \mathbf{b}) + \hat{d}(\mathbf{b}, \mathbf{c}) \geq \hat{d}(\mathbf{a}, \mathbf{c})$, *for any* $\mathbf{a}, \mathbf{b}, \mathbf{c} \in \mathbb{R}$.

*Proof.* To prove this lemma, it is equivalent to prove the following inequality by definition of $\hat{d}$,

$$
\sqrt{\|\mathbf{a}\|^2 + \|\mathbf{b}\|^2 - \mathbf{a}^\top \mathbf{b}} + \sqrt{\|\mathbf{b}\|^2 + \|\mathbf{c}\|^2 - \mathbf{b}^\top \mathbf{c}} \geq \sqrt{\|\mathbf{a}\|^2 + \|\mathbf{c}\|^2 - \mathbf{a}^\top \mathbf{c}}.
\tag{15}
$$

Because $-\|\mathbf{x}\|\|\mathbf{y}\| \leq \mathbf{x}^\top \mathbf{y} \leq \|\mathbf{x}\|\|\mathbf{y}\|, \forall \mathbf{x}, \mathbf{y}$, we have

$$
\begin{aligned}
&\sqrt{\|\mathbf{a}\|^2 + \|\mathbf{b}\|^2 - \mathbf{a}^\top \mathbf{b}} + \sqrt{\|\mathbf{b}\|^2 + \|\mathbf{c}\|^2 - \mathbf{b}^\top \mathbf{c}} \\
\geq & \sqrt{\|\mathbf{a}\|^2 + \|\mathbf{b}\|^2 - \|\mathbf{a}\|\|\mathbf{b}\|} + \sqrt{\|\mathbf{b}\|^2 + \|\mathbf{c}\|^2 - \|\mathbf{b}\|\|\mathbf{c}\|},
\end{aligned}
\tag{16}
$$

and

$$
\sqrt{\|\mathbf{a}\|^2 + \|\mathbf{c}\|^2 + \|\mathbf{a}\|\|\mathbf{c}\|} \geq \sqrt{\|\mathbf{a}\|^2 + \|\mathbf{c}\|^2 - \mathbf{a}^\top \mathbf{c}}.
\tag{17}
$$

If

$$
\sqrt{\|\mathbf{a}\|^2 + \|\mathbf{b}\|^2 - \|\mathbf{a}\|\|\mathbf{b}\|} + \sqrt{\|\mathbf{b}\|^2 + \|\mathbf{c}\|^2 - \|\mathbf{b}\|\|\mathbf{c}\|} \geq \sqrt{\|\mathbf{a}\|^2 + \|\mathbf{c}\|^2 + \|\mathbf{a}\|\|\mathbf{c}\|}
\tag{18}
$$

is true, then inequality (15) is true and Lemma A.4 is proven. To prove inequality (18), we can take squares on both sides without sign changing because both sides are non-negative. Then we have,

$$
\left( \sqrt{\|\mathbf{a}\|^2 + \|\mathbf{b}\|^2 - \|\mathbf{a}\|\|\mathbf{b}\|} + \sqrt{\|\mathbf{b}\|^2 + \|\mathbf{c}\|^2 - \|\mathbf{b}\|\|\mathbf{c}\|} \right)^2 \geq \|\mathbf{a}\|^2 + \|\mathbf{c}\|^2 + \|\mathbf{a}\|\|\mathbf{c}\|.
\tag{19}
$$

To prove inequality (18), it is equivalent to prove inequality (19). Expand and simplify inequality (19), we have

$$
2\sqrt{\|\mathbf{a}\|^2 + \|\mathbf{b}\|^2 - \|\mathbf{a}\|\|\mathbf{b}\|} \sqrt{\|\mathbf{b}\|^2 + \|\mathbf{c}\|^2 - \|\mathbf{b}\|\|\mathbf{c}\|} \geq -2\|\mathbf{b}\|^2 + \|\mathbf{a}\|\|\mathbf{b}\| + \|\mathbf{b}\|\|\mathbf{c}\| + \|\mathbf{a}\|\|\mathbf{c}\|.
\tag{20}
$$

The left-hand side of inequality (20) is non-negative.

1) if right hand side $-2\|\mathbf{b}\|^2 + \|\mathbf{a}\|\|\mathbf{b}\| + \|\mathbf{b}\|\|\mathbf{c}\| + \|\mathbf{a}\|\|\mathbf{c}\| < 0$, then inequality (20) is proven and backtrace to Lemma A.4 is proven.

2) if right hand side $-2\|\mathbf{b}\|^2 + \|\mathbf{a}\|\|\mathbf{b}\| + \|\mathbf{b}\|\|\mathbf{c}\| + \|\mathbf{a}\|\|\mathbf{c}\| \geq 0$, we take square on both sides of inequality (20) and have,

$$4(\|\mathbf{a}\|^2 + \|\mathbf{b}\|^2 - \|\mathbf{a}\|\|\mathbf{b}\|)(\|\mathbf{b}\|^2 + \|\mathbf{c}\|^2 - \|\mathbf{b}\|\|\mathbf{c}\|) \geq (-2\|\mathbf{b}\|^2 + \|\mathbf{a}\|\|\mathbf{b}\| + \|\mathbf{b}\|\|\mathbf{c}\| + \|\mathbf{a}\|\|\mathbf{c}\|)^2. \tag{21}$$

To prove inequality (21), we let left-hand side subtract right-hand side,

$$\begin{aligned}
&4(\|\mathbf{a}\|^2 + \|\mathbf{b}\|^2 - \|\mathbf{a}\|\|\mathbf{b}\|)(\|\mathbf{b}\|^2 + \|\mathbf{c}\|^2 - \|\mathbf{b}\|\|\mathbf{c}\|) - (-2\|\mathbf{b}\|^2 + \|\mathbf{a}\|\|\mathbf{b}\| + \|\mathbf{b}\|\|\mathbf{c}\| + \|\mathbf{a}\|\|\mathbf{c}\|)^2 \\
=&3\|\mathbf{a}\|^2\|\mathbf{b}\|^2 + 3\|\mathbf{a}\|^2\|\mathbf{c}\|^2 + 3\|\mathbf{b}\|^2\|\mathbf{c}\|^2 - 6\|\mathbf{a}\|^2\|\mathbf{b}\|\|\mathbf{c}\| - 6\|\mathbf{a}\|\|\mathbf{b}\|\|\mathbf{c}\|^2 + 6\|\mathbf{a}\|\|\mathbf{b}\|^2\|\mathbf{c}\| \\
=&3(\|\mathbf{a}\|\|\mathbf{b}\| + \|\mathbf{b}\|\|\mathbf{c}\| - \|\mathbf{a}\|\|\mathbf{c}\|)^2 \\
\geq&0.
\end{aligned} \tag{22}$$

Therefore, inequality (21) is proven. Consequently, inequality (20) in the case of $-2\|\mathbf{b}\|^2 + \|\mathbf{a}\|\|\mathbf{b}\| + \|\mathbf{b}\|\|\mathbf{c}\| + \|\mathbf{a}\|\|\mathbf{c}\| \geq 0$ is proven. Summarize with the case of $-2\|\mathbf{b}\|^2 + \|\mathbf{a}\|\|\mathbf{b}\| + \|\mathbf{b}\|\|\mathbf{c}\| + \|\mathbf{a}\|\|\mathbf{c}\| < 0$, inequality (20) is proven and Lemma A.4 is proven.

$\square$

**Theorem A.5.** $\hat{d}$ *is a diffuse metric.*

*Proof.* By summarizing Lemma A.2, A.3, and A.4, function $\hat{d}$ holds the three axioms of diffuse metric. Therefore, function $\hat{d}$ is a diffuse metric. $\square$

### A.3.3  Proof of Theorem 3.6

**Theorem 5.** *Let $\mathbb{M}_\psi$ be the space of $d_\psi$. The metric update operator $\mathcal{F}_{Chrono} : \mathbb{M}_\psi \to \mathbb{M}_\psi$ is defined as,*

$$\mathcal{F}_{Chrono} d_\psi(\mathbf{x}_i, \mathbf{x}_j, \mathbf{y}_{i'}, \mathbf{y}_{j'}) = |r_{\mathbf{x}_i} - r_{\mathbf{y}_{i'}}| + \gamma \mathbb{E}_{\mathbf{x}_{i+1} \sim P_\mathbf{x}^\pi, \mathbf{y}_{i'+1} \sim P_\mathbf{y}^\pi} d_\psi(\mathbf{x}_{i+1}, \mathbf{x}_j, \mathbf{y}_{i'+1}, \mathbf{y}_{j'}). \tag{23}$$

*$\mathcal{F}_{Chrono}$ has a fixed point.*

*Proof.* In practice, we have assumed that the sampled $\mathbf{x}_j$ and $\mathbf{y}_{j'}$ are future states of $\mathbf{x}_i$ and $\mathbf{y}_{i'}$, respectively. To be precise, we extend the definition of $\mathcal{F}_{Chrono}$ to cover the cases that $i > j$ or $i' > j'$.

$$\mathcal{F}_{Chrono} d_\psi(\mathbf{x}_i, \mathbf{x}_j, \mathbf{y}_{i'}, \mathbf{y}_{j'}) = \begin{cases} 0, & i > j \text{ or } i' > j', \\ |r_{\mathbf{x}_i} - r_{\mathbf{y}_{i'}}| + \gamma \mathbb{E}_{\mathbf{x}_{i+1} \sim P_\mathbf{x}^\pi, \mathbf{y}_{i'+1} \sim P_\mathbf{y}^\pi} d_\psi(\mathbf{x}_{i+1}, \mathbf{x}_j, \mathbf{y}_{i'+1}, \mathbf{y}_{j'}), & \text{otherwise.} \end{cases} \tag{24}$$

We follow the proof in Section A.3.1. $\mathcal{F}_{Chrono}$ is contraction mapping because.

$$\begin{aligned}
&|\mathcal{F}_{Chrono} d_\psi(\mathbf{x}_i, \mathbf{x}_j, \mathbf{y}_{i'}, \mathbf{y}_{j'}) - \mathcal{F}_{Chrono} d_{\psi'}(\mathbf{x}_i, \mathbf{x}_j, \mathbf{y}_{i'}, \mathbf{y}_{j'})| \\
=&\left| \gamma \mathbb{E}_{\mathbf{x}_{i+1} \sim P_\mathbf{x}^\pi, \mathbf{y}_{i'+1} \sim P_\mathbf{y}^\pi} (d_\psi - d_{\psi'})(\mathbf{x}_{i+1}, \mathbf{x}_j, \mathbf{y}_{i'+1}, \mathbf{y}_{j'}) \right| \\
\geq&\gamma \|(d_\psi - d_{\psi'})(\mathbf{x}_{i+1}, \mathbf{x}_j, \mathbf{y}_{i'+1}, \mathbf{y}_{j'})\|_\infty
\end{aligned} \tag{25}$$

By Banach's fixed point theorem, operator $\mathcal{F}_{Chrono}$ has a fixed point $d_\psi^\pi$. $\square$

### A.4  Theoretical Details

### A.4.1  Łukaszyk-Karmowski Distance

**Definition A.6** (Łukaszyk-Karmowski Distance). Given a metric space $(X, d)$ where $d$ is a metric defined on $X \times X$, the Łukaszyk-Karmowski distance [49] is a distance between two probability measures $\mu$ and $\nu$ on $X$ using metric $d$, and is defined as follows:

$$d_{LK}(d)(\mu, \nu) = \mathbb{E}_{x \sim \mu, x' \sim \nu} d(x, x') \tag{26}$$

Intuitively, $d_{LK}$ measures the expected distance between two samples drawn from $\mu$ and $\nu$, respectively. If we measure the $d_{LK}(d)$ distance between two identical distributions, the distance is not restricted to zero, i.e., $d_{LK}(d)(\mu, \mu) \geq 0$.

### A.5 Implementation Details

#### A.5.1 Implementation of $\hat{m}$

We adopt IQE [38] to implement $\hat{m}$. Given two vectors $\mathbf{a}, \mathbf{b} \in \mathbb{R}^n$, reshaping to $\mathbb{R}^{k \times l}$ where $k \times l = n$, IQE first computes the union of the interval for each component:

$$d_i(\mathbf{a}, \mathbf{b}) = | \bigcup_{j=1}^{l} [\mathbf{a}_{ij}, \max(\mathbf{a}_{ij}, \mathbf{b}_{ij})] |, \forall i = 1, 2, ..., k, \tag{27}$$

where $[\cdot, \cdot]$ is the interval on the real line. Then it computes the distance among all components $d_i$ as

$$d_{IQE}(\mathbf{a}, \mathbf{b}) = \alpha \cdot \max(d_1(\mathbf{a}, \mathbf{b}), ..d_k(\mathbf{a}, \mathbf{b})) + (1 - \alpha) \cdot \text{mean}(d_1(\mathbf{a}, \mathbf{b}), ..d_k(\mathbf{a}, \mathbf{b})), \tag{28}$$

where $\alpha \in \mathbb{R}$ is an adaptive weight to balance the "max" and "mean" terms. In the scope of our method, we adopt $d_{IQE}(\mathbf{a}, \mathbf{b})$ to implement $\hat{m}$.

#### A.5.2 Network Architecture

The encoder $\phi$ takes an input of the states and consists of 4 convolutional layers followed by 1 fully-connected layer. The output dimension of $\phi$ is 256. The encoder $\psi$ takes an input of 512-dimensional vector (concatenated with $\phi(\mathbf{x}_i)$ and $\phi(\mathbf{x}_j)$), feed it into two layer MLPs with 512 hidden units, and output a 256 dim embedding. Q network and policy network are 3-layer MLPs with 1024 hidden units.

#### A.5.3 Hyperparameters

| Hyperparameters | Values |
|---|---|
| Stack frames | 3 |
| Observation shape | $(3 \times 3, 84, 84)$ |
| Action repeat | 2 for finger-spin, walker-walk |
| | 8 for cartpole-swing_up, cartpole-swing_up_sparse |
| | 4 for others in DeepMind Control Suite |
| | 1 for Meta-World |
| Convolutional layers | 4 |
| Convolutional kernal size | $3 \times 3$ |
| Convolutional strides | $[2, 1, 1, 1]$ |
| Convolutional channels | 32 |
| $\phi$ dimension | 256 |
| $\psi$ dimension | 256 |
| Learning rate | 1e-4 |
| Q function EMA $\alpha_Q$ | 0.01 |
| Encoder $\phi$ EMA $\alpha_\phi$ | 0.05 |
| Initial steps | 1000 |
| Replay buffer size | 500K |
| Target update freq | 2 |
| Batch size | 128 |
| Discount factor $\gamma$ | 0.95 for ball_in_cup-catch, Reacher-easy |
| | 0.99 for others |

Table 4: Hyperparameters

### A.5.4 Algorithm

---

**Algorithm 1** A learning step in jointly learning `SCR` and SAC.

---

**Require:** Replay Buffer $\mathcal{D}$, Q network $Q$, policy network $\pi$, target Q network $\bar{Q}$, state encoder $\phi$, target state encoder $\bar{\phi}$, chronological encoder $\psi$.

Sample a batch of trajectories with size $B$: $\{\tau_k\}_{k=1}^{B} \sim \mathcal{D}$

Sample state $\mathbf{x}_i$, transition at $\mathbf{x}_i$ and its future state $\mathbf{x}_j$ from each trajectory $\tau_k$: $\{(\mathbf{x}_i, \mathbf{x}_{i+1}, r_i, \mathbf{a}_i, \mathbf{x}_j)_k \sim \tau_k\}_{k=1}^{B}$

Compute loss $\mathcal{L}_{\phi}(\phi)$ in (2)

Compute loss $\mathcal{L}_{\psi}(\psi, \phi)$ in (5)

Compute loss $\mathcal{L}_{\hat{m}}(\phi)$ in (9)

Compute loss $\mathcal{L}(\phi, \psi) = \mathcal{L}_{\phi}(\phi) + \mathcal{L}_{\psi}(\psi, \phi) + \mathcal{L}_{\hat{m}}(\phi)$ in (10)

Update $\phi$ and $\psi$ by minimizing loss $\mathcal{L}(\phi, \psi)$

Compute RL loss $\mathcal{L}_{RL}$ according to SAC objectives

Update $\phi$, $Q$ and $\pi$ by minimizing loss $\mathcal{L}_{RL}$

Soft update target Q network: $\bar{Q} = \alpha_Q Q + (1 - \alpha_Q)\bar{Q}$

Soft update target state encoder: $\bar{\phi} = \alpha_{\phi}\phi + (1 - \alpha_{\phi})\bar{\phi}$

---

```python
def d(x, y):
    a = (x.pow(2.).sum(dim=-1, keepdim=True) + y.pow(2.).sum(dim=-1,
    keepdim=True)) # x^2 + y^2
    b = (x * y).sum(dim=-1, keepdim=True) # xy
    return (a - b).sqrt() # sqrt(x^2 + y^2 - xy)

def encoder_loss(self, s_i, s_i_1, s_j, s_j_1, r_i, r_j, agg_rew_i_j):
    phi_i = self.encoder(s_i)
    phi_i_1 = self.encoder(s_i_1)
    phi_j = self.encoder(s_j)
    phi_j_1 = self.encoder(s_j_1)
    # concatenate i and j for compute loss (2) where j is not required
    phi_t = torch.cat([phi_i, phi_j])
    phi_t1 = torch.cat([phi_i_1, phi_j_1])
    r_t = torch.cat([r_i, r_j])
    # compute loss (2)
    # permute batch to create y
    perm = torch.randperm(phi_t.shape[0])
    d_phi_t = d(phi_t, phi_t[perm])
    d_phi_t1 = d(phi_t1, phi_t1[perm])
    # loss (2)
    loss_phi = F.mse_loss(d_phi_t, r_t + self.discount * d_phi_t1.
    detach())

    # compute loss (5)
    psi = self.psi_mlp(phi_i, phi_j)
    psi_1 = self.psi_mlp(phi_i_1, phi_j)
    # permute batch to create y
    perm = torch.randperm(psi.shape[0])
    d_psi = d(psi, psi[perm])
    d_psi_1 = d(psi_1, psi_1[perm])
    # loss (5)
    loss_psi = F.mes_loss(d_psi, r_i + self.discount * d_psi_1.detach
    ())

    # compute m
    m = iqe(phi_i, phi_j)
    # compute loss (8)
    low_bound = agg_rew_i_j
    loss_low = (F.mse_loss(m, low_bound, reduction='none') * (m <
    low_bound).detach()).mean()
    # compute loss (10)
```

```
39    up_bound = d(phi_i, phi_i[perm]) + d(phi_j, phi_j[perm]) +
      agg_rew_i_j[perm]
40    up_bound = up_bound.detach()
41    loss_up = (F.mse_loss(m.abs(), up_bound, reduction='none') * (m.
      abs() > up_bound).detach()).mean()
42
43    return loss_phi + loss_psi + loss_low + loss_up
44
```

Listing 1: Pseudo code of learning of representation

## A.6 Computational Resources

The experiment is done on servers with an 128-cores CPU, 512 GB RAM and NVIDIA A100 GPUs. Two instances are running in one GPU at the same time.

# B Experiments

## B.1 Additional Information for Distraction Settings in DeepMind Control Suite

For the distraction setting in Section 4.2, we utilized the Distracting Control Suite [34] with the setting "difficulty=easy". This involves mixing the background with videos from the DAVIS2017 [29] dataset. Specifically, the training environment samples videos from the DAVIS2017 train set, while the evaluation environment uses videos from the validation set. Each episode reset triggers the sampling of a new video. Additionally, it introduces variability in each episode by applying a uniformly sampled RGB color shift to the robot's body color and randomly selecting the camera pose. The specifics of the RGB color shift range and camera pose variations are in line with the Distracting Control Suite paper [34]. Different random seeds are used for the training and evaluating environments at the start of training to ensure diverse environments.

## B.2 Name of Tasks in DeepMind Control Suite

Due to limited space in main text, we use short name for tasks in DeepMind Control Suite. The full name of tasks are listed in below Table 5:

| Short Name | Full Name |
|---|---|
| BiC-Catch | ball_in_cup-catch |
| C-SwingUp | cartpole-swing_up |
| C-SwingUpSparse | cartpole-swing_up_sparse |
| Ch-Run | cheetah-run |
| F-Spin | finger-spin |
| H-Stand | hopper-stand |
| R-Easy | Reacher-easy |
| W-Walk | walker-walk |

Table 5: Full name of tasks in DeepMind Control Suite

## B.3 Additional Results of Distracting Control Suite in Section 4.2

We extend the experiments in Section 4.2 with compared with RAP [8], showing results in Table 6. Figure 9 shows the training curves of SCR and baseline methods in distraction setting of DM_Control.

|      | BiC-Catch | C-SwingUp | C-SwingUpSparse | Ch-Run | F-Spin | H-Stand | R-Easy | W-Walk |
|------|-----------|-----------|-----------------|--------|--------|---------|--------|--------|
| SAC  | 82.6±20.2 | 218.4±4.9 | 0.7±0.7 | 177.4±8.4 | 22.5±27.7 | 19.6±26.2 | 149.4±77.5 | 167.2±8 |
| DrQ  | 124.0±99.6 | 230.0±36.3 | 11.2±6.4 | 103.0±84.9 | 579.8±282.8 | 16.8±11.8 | 70.5±40.1 | 33.6±6.3 |
| DBC  | 48.8±24.4 | 127.7±19.2 | 0.0±0.0 | 9.2±1.9 | 7.7±10.7 | 5.6±2.5 | 149.6±42.8 | 30.9±4.8 |
| MICo | 104.2±10 | 200.2±6.6 | 0.0±0.1 | 7.4±3.2 | 86.9±76.1 | 11.8±10.5 | 132.3±37.8 | 27.5±7.7 |
| SimSR | 106.4±13.5 | 148.4±17.3 | 0.0±0.0 | 28.2±23.8 | 0.4±0.1 | 6.6±1.2 | 78.4±17 | 28.6±3.1 |
| RAP  | 82.6±159.3 | 451.7±74.1 | 0.1±0.7 | 243.2±26.4 | 577.7±140.4 | 6.1±11.7 | 87.8±86.7 | 183.7±108.7 |
| SCR  | **221.3**±55.4 | **565.1**±59.9 | **185.7**±93 | **331.7**±1.4 | **738.3**±24.5 | **400.8**±19.2 | **666.1**±14.5 | **555.1**±31.2 |

Table 6: Results (mean±std) on DM_Control with distraction setting at 500K environment step. Distraction includes background, robot body color, and camera pose.

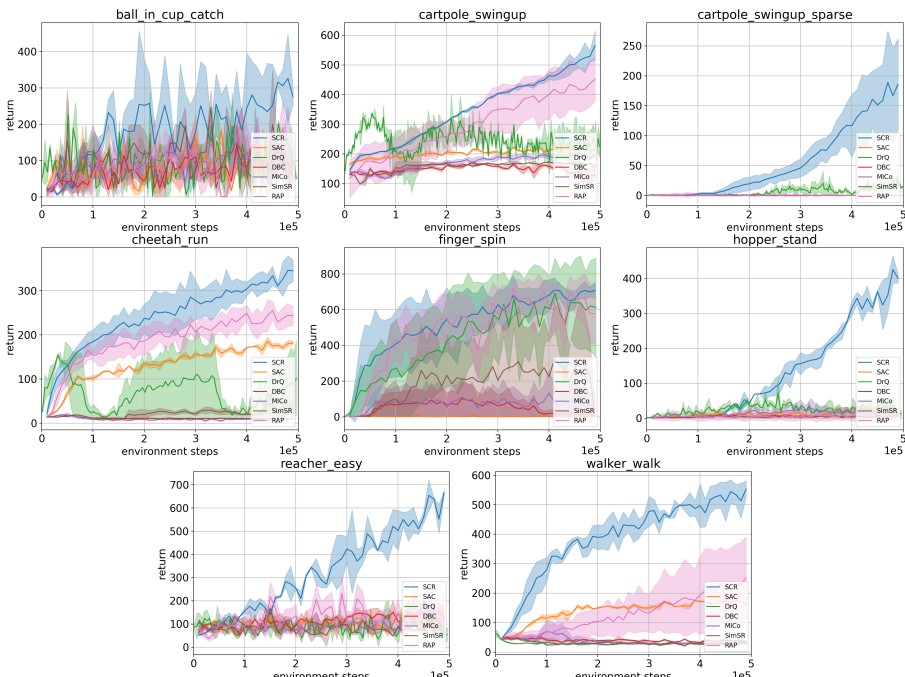

Figure 9: Training curves of SCR and baseline methods in distraction setting of DM_Control. Curves are evaluation scores average on 10 runs and shadow shapes are std.

## B.4 Additional Comparison Experiments with Data Augmentation Methods

Data augmentation is a category of methods efficient on reinforcement learning, toward sample efficiency and noise invariance. Additional baseline CBM [26] is bisimulation metric method combing with data augmentation method DrQ-v2, and it achieves significant performance in distracting DM_Control. To compare with CBM, we embed data augmentation method DrQ into our proposed method SCR. Table 7 indicates that the performance of SCR combined with DrQ surpass baseline CBM.

|      | BiC-Catch | C-SwingUp | C-SwingUpSparse | Ch-Run | F-Spin | H-Stand | R-Easy | W-Walk |
|------|-----------|-----------|-----------------|--------|--------|---------|--------|--------|
| DrQ  | 124.0±99.6 | 230.0±36.3 | 11.2±6.4 | 103.0±84.9 | 579.8±282.8 | 16.8±11.8 | 70.5±40.1 | 33.6±6.3 |
| CBM  | 642.5±66.1 | 464.1±85.4 | 27.6±5.2 | 422.1±82.0 | **817.6**±21.7 | 212.9±24.6 | 319.5±274.6 | 688.5±65.6 |
| SCR  | 221.3±55.4 | 565.1±59.9 | 185.7±93.0 | 331.7±1.4 | 738.3±24.5 | 400.8±19.2 | 666.1±14.5 | 555.1±31.2 |
| SCR w/ DrQ | **936.0**±97.0 | **783.9**±62.0 | **417.2**±89.5 | **480.2**±48.2 | 835.3±37.3 | **706.4**±165.5 | **829.0**±32.3 | **835.6**±79.5 |

Table 7: Results (mean±std) on DM_Control with distraction setting at 500K environment step. Distraction includes background, robot body color, and camera pose.

## B.5 Training Curves Meta-World

Figure 10 shows the training curves of SCR and baseline methods in Meta-World in Section 4.4.

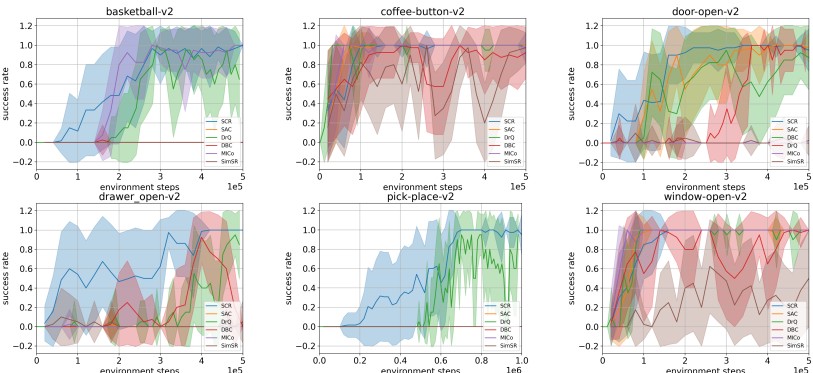

Figure 10: Training Curves of Meta-World. Mean success rates on 5 runs with std (shadow shape).

## B.6 Additional Experiments on MiniGrid

We provide additional experiments on MiniGrid-FourRooms [9] environment. MiniGrid is under a sparse reward setting that the agent receives a reward only when it successfully reaches a specific goal, without rewards at previous steps. This experiment verifies the motivation of SCR that is able to handle non-informative rewards. We compare various metric-based representation methods that combine with backbone PPO [31] algorithms. We train each run for 5M environment steps. Table shows the results with PPO, PPO+DBC, PPO+MICo, PPO+SimSR, and PPO+SCR.

| Methods | PPO | PPO+DBC | PPO+MICo | PPO+SimSR | PPO+SCR |
|---|---|---|---|---|---|
| scores | 0.515 ±0.029 | 0.515 ±0.012 | 0.350 ±0.064 | 0.321 ±0.046 | **0.546** ±0.012 |

Table 8: Results (mean±std) on MiniGrid-FourRooms [9] at 5M environment steps. Results are average over 5 seeds.

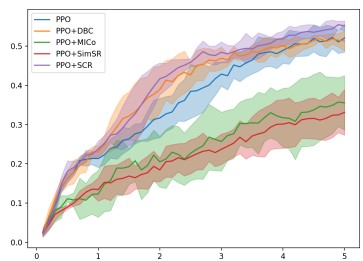

Figure 11: Training curves on MiniGrid-FourRooms [9]. Results are average over 5 seeds.

