# OpenReview forum: "State Chrono Representation for Enhancing Generalization in Reinforcement Learning"
_NeurIPS.cc/2024/Conference — NeurIPS 2024 poster_

### Official Review · Reviewer_UVpA · 2024-06-30

**Soundness:** 2
**Presentation:** 3
**Contribution:** 3
**Rating:** 5
**Confidence:** 4

**Summary:**

The paper proposes an algorithm to improve upon the recently proposed class of deep bisimulation-based methods by accounting for long-term consequences (in terms of future states) instead of relying one-step bootstrapping based on reward differences to be more robust in settings such as sparse reward cases.

**Strengths:**

1. I think the idea is valuable. The sparse reward setting is challenging for bisimulation-based methods and this type of idea of factoring future states is a reasonable attempt to make these algorithms better.
2. The algorithm is evaluated on a variety of challenging pixel-based environments, which is good when evaluating representation learning algorithms.

**Weaknesses:**

1. It is a bit concerning that the main results are over only 5 seeds. Personally, it makes it hard for me to reach a conclusion on the performance because of this. I’d refer authors to [1].
2. Related to above, the variance of the reported results are quite high. For example, in Table 1, 2 and Figure 3, the variance is so high that it is difficult to conclusively say which algorithm is better than another.
3. Table 3 does not report the number of trials nor any measure of variance.
4. While the approach proposed for bisimulation methods is indeed unique, I believe there are strong conceptual relations to self-predictive representation algorithms and successor representations [2, 3]. It would be good if these were discussed in the related works section and if similarities/differences can be drawn between SCR and these related works to better shape the work and help readers.
5. It appears like Section 3.1 is basically prior work, but the paper includes it as though it is a contribution. I think it would be better to move this to the Appendix since the actual contributions are the sections after Section 3.1.

If the above issues are adequately addressed, I will consider increasing the score.

[1] Empirical Design in Reinforcement Learning. Patterson et al. 2023.

[2] Data-Efficient Reinforcement Learning with Self-Predictive Representations. Schwarzer et al. 2020.

[3] Understanding Self-Predictive Learning for Reinforcement Learning. Tang et al. 2022

**Questions:**

1. A common issue in self-predictive representation algorithms (see Weaknesses above) is that the training procedure may lead to representation collapse [1]. Is SCR susceptible to this as well? It appears like it may be so. If not, how is that avoided here?
2. I am curious how much of a burden it is to track future states in a given trajectory? A benefit of the other deep bismulation methods is that they can operate on a dataset of the form $\{(s,a’,s’)\}$, which is very useful in off-policy learning. SCR seems to require trajectory knowledge which may be a memory burden. In general, this is one of the advantages of one-step bootstrapping over n-step returns. If this is a limitation, it should be mentioned in the limitations section.
3. Related to the above, how far into the future does a state $y$ have to be from previous state $x$ when computing the loss function. That is, using the terminology of n-step returns, how many steps in the future does $y$ have to be when computing these loss functions?


[1] Understanding Self-Predictive Learning for Reinforcement Learning. Tang et al. 2022

**Limitations:**

Yes, the paper addresses this in Section 5.

---

> ### Author Rebuttal · Authors · 2024-08-07
>
> Thanks for your effort and insightful comments.
>
> > Weakness 1.& 2.
>
> We appreciate your concern regarding the number of seeds. Due to resource constraints, we opted for a balance between the number of seeds and computational cost, thus using 5 seeds. To enhance the robustness of our results, each seed was evaluated over 100 episodes, totaling **500(=5x100) episodes** across all seeds. The mean and std scores in **Table 1** and **2** are calculated over all 500 returns.
>
> The high variance in Table 2 come from the internal randomness of the distraction settings in the DM_Control environment, where backgrounds, body colors, and camera poses vary per episode. The combination of these attributes leads to varying distraction effects, making some episodes more challenging and resulting in lower scores. This variance underscores the challenges of these settings and validates the necessity of our approach for such distraction setting.
>
> To further ensure the reliability of the results, we present **aggregate metrics** in Figure 4, a common practice to compare different RL algorithms. Figure 4 demonstrates that our SCR significantly outperforms other baselines in the **distraction setting**.
>
> Regarding the results in the **default setting** shown in **Table 1**, our intention is not to claim that SCR outperforms other baselines but to demonstrate that it is **comparable** to them (**lines 206-308**). The default setting of DM_Control has been extensively studied, with SOTA methods achieving high scores. The purpose of including results in the default setting is to verify SCR's effectiveness in a standard scenario. Combined with the results in Table 2, this further confirms that SCR **enhances generalization** rather than merely **trading off** performance between distraction and default settings. Table 1 shows that the mean scores of SCR are **comparable to baseline methods**, with **variance at a similar level**, thereby validating our claims.
>
> >3.
>
> Table 3 is also evaluated over 5 seeds, with each seed evaluated over 100 episodes. We update the table to include standard deviations.
> |SAC |DrQ| DBC| MICo| SimSR |SCR|
> |-|-|-|-|-|-|
> |0.495 $\pm$ 0.475|0.886 $\pm$ 0.125|0.479 $\pm$ 0.453|0.495 $\pm$ 0.482|0.258 $\pm$ 0.365|0.969 $\pm$ 0.032|
>
> >4.
>
> Self-predictive representations (SPR) shares similarities with SCR, as both methods learn representations regarding a future state. However, there are several key differences between them. SPR focuses on **predicting future states**, enforcing the representation of predicted future states to be close to the true future states. In contrast, SCR focuses on **predicting a metric/distance** between current state and future state. The metrics in SCR (in Section 3.1-3.3) are all related to rewards or values, which is correlated to policy learning. SPR's representation focuses on dynamics learning, without directly considering the reward or value.
>
> Successor Representations (SRs) are a class of methods designed to learn representations that facilitate generalization. SRs achieve this by focusing on the **occupancy of states**, enabling generalization across various tasks and reward functions. In contrast, SCR measures the distance between states specifically to handle observable distractions.
>
> >5.
>
> We would like to clarify the contributions of Section 3.1. The major **technical contributions** are as follows:
> 1. We identify the limitations of existing metric approximations.
> 2. We propose a novel approximation metric, detailed in Definition 3.3.
> 3. We provide theoretical support for this new metric in Theorem 3.4 and Lemma 3.5.
>
> These points are original contributions of our paper. Empirical results in the ablation study (**Section 4.3 / Figure 5**) show that substituting our proposed metric with existing ones like MICo, SimSR, or DBC results in performance degradation, highlighting the effectiveness of our approach in Section 3.1.
>
> From a writing perspective, retaining Section 3.1.1 in the main text is important as it introduces the long-term temporal information issues that motivate the subsequent sections (3.2 and 3.3). Although Eq.1 may appear less novel due to its similar form to existing work, it sets the stage for the necessity of designing a new approximation metric and leads to Eq.2, which is integral to the final objective (Eq.10) of SCR. We will revise Section 3.1 to enhance clarity and explicitly highlight its contributions.
>
> > Questions 1.
>
> Representation collapse in bisimulation metrics has been discussed in [1]. This issue can arise in scenarios with sparse rewards, where the reward difference term in Eq.1 is always zero, leading to the collapse of the distance between states $x$ and $y$. Addressing sparse rewards is one of the objectives of our paper.
>
> To mitigate the risk of collapse, we introduce a temporal measurement that extend to multiple states, thereby increasing the likelihood of encountering non-zero rewards. Additionally, the representation function in SCR is continuously updated through the RL objectives, which provide additional gradients and help avoiding the representation collapse.
>
> >2.
>
> We store entire episodes in the replay buffer in a contiguous manner, with each step stored as $(s, a, s')$, similar to other one-step off-policy methods. The only additional memory burden is the storage of indices for the start and end of each episode. These indices are integers, therefore, the memory overhead is negligible.
>
> When sampling a batch of training data from the replay buffer, we first uniformly sample a batch of steps $i$. We then sample steps $j$ based on the indices of the start and end of the episode, ensuring that steps $i$ and $j$ are within the same episode. Finally, we swap $i$ and $j$ if $j > i$ to ensure that $j$ is a future step relative to $i$. This method keeps the memory overhead minimal while effectively tracking future states.

---

> > ### Author Response · Authors · 2024-08-07
> > **Rebuttal by Authors (Continued)**
> >
> > >3.
> >
> > The parameter $n$ is uniformly sampled within the range [1, 100], as described in **lines 338-344** of the manuscript. Our ablation study, detailed in **Section 4.3 / Figure 6**, examines the impact of different $n$. The results indicate that sampling $n$ within the [1, 100] range provides optimal performance and maintains stability across various tasks.
> >
> > ----
> > [1] Towards Robust Bisimulation Metric Learning. Kemertas et al. NeurIPS 2021.

---

> > > ### Comment · Reviewer_UVpA · 2024-08-08
> > >
> > > Thanks to the authors for their response. Two questions:
> > > 1. Re: seeds. My understanding of how this typically works is: 1 algorithm on 1 environment with 1 configuration of h-params is run for $N$ seeds. These $N$ results are averaged together. In my opinion, the real robustness improves by not increasing the number of episodes but by increasing the number of seeds. I see computational constraint as an issue, but I don't think doing 5 to 10 more seeds is that expensive.
> > >
> > > 2. Do the aggregate metrics report an IQM? If so, I don't think that is the best reporting especially given that your response mentions that the distraction setting results in high variance. If the setup, by nature, is high variance, then I don't think we want to exclude the extreme values if the goal is to say the algorithm performs well in high-variance settings.

---

> > > > ### Author Response · Authors · 2024-08-10
> > > > **Response to Reviewer UVpA**
> > > >
> > > > > Re: seeds. My understanding of how this typically works is: 1 algorithm on 1 environment with 1 configuration of h-params is run for $N$ seeds. These $N$ results are averaged together. In my opinion, the real robustness improves by not increasing the number of episodes but by increasing the number of seeds.
> > > >
> > > > We agree with you a comprehensive comparison is needed for readers to better understand our SCR capabilities. In this comment, we show the results with **std calculated over 5 averaged scores** in the following tables, ***Table A and B***. The std is much lower than the tables in the manuscript.
> > > >
> > > > We did not show the std calculated over $N$ averaged scores in the manuscript, because the averaged scores **decline the effect of the extreme values in final std** and conceal the high variance issue of the challenging tasks. Instead, we show the std calculated over all $N \times num\\_episodes$ scores because it can reflect the **extreme values**.
> > > >
> > > > **A large number of evaluation episodes is necessary** to improve the confidence level of the average scores, due to the high variance across various episodes. The high variance is from the large randomness of the Distracting DM_Control environment. It randomly selects a background video, a body color, and a camera pose at the beginning of each episode. Different episodes will have different distraction effects, which consequently affect the agent's performance.
> > > >
> > > >
> > > > > I see computational constraint as an issue, but I don't think doing 5 to 10 more seeds is that expensive.
> > > >
> > > > We conduct experiments by running each instance on 0.5 GPU for 12 hours. 0.5 GPU means we run 2 instances in parallel on 1 GPU. For the DM_Control experiment, we have 6 algorithms, 8 tasks, and 2 configurations (default and distraction) for 5 seeds. In total, we need **6 * 8 * 2 * 5 * 0.5 * 12 = 2880 GPU hours**. We understand that more seeds would lead to more robust results. However, due to computational constraints, we don't believe we can train 5 to 10 more seeds during the rebuttal period.
> > > >
> > > > 5 seeds is a common practice in baseline papers, such as SimSR and MICo. We follow the same practice to ensure a fair comparison with the baselines. Increasing the number of seeds can improve confidence, which is beneficial for this paper, and we **will increase the number of seeds in the revised version.**
> > > >
> > > > The results in ***Table 2 and Figure 4*** in the manuscript and ***Table B*** in this comment show that the performance of SCR is **significantly better** than the baselines in the distraction setting. We believe these result sufficiently prove the effectiveness of SCR.
> > > >
> > > >
> > > > > Do the aggregate metrics report an IQM? If so, I don't think that is the best reporting especially given that your response mentions that the distraction setting results in high variance. If the setup, by nature, is high variance, then I don't think we want to exclude the extreme values if the goal is to say the algorithm performs well in high-variance settings.
> > > >
> > > > The aggregate metrics in **Figure 4** include median, IQM, mean, and optimality gap. The mean and optimality gap metrics do not exclude the extreme values.
> > > > Your concern regarding the metrics is reasonable. However, the community has **no perfect metric** to measure performance in high-variance environments currently. To mitigate the potential bias, we show the std over all evaluation episodes which can **include the extreme values**.

---

> ### Author Response · Authors · 2024-08-10
> **Response to Reviewer UVpA (Continued)**
>
> Methods | BiC-Catch | C-SwingUp | C-SwingUpSparse | Ch-Run | F-Spin | H-Stand | R-Easy | W-Walk
> ---|---|---|---|---|---|---|---|---
> SAC | 465.9 $\pm$ 12.7 | 730.2 $\pm$ 101.9 | 21.1 $\pm$ 10.6 | 357.5 $\pm$ 3.4 | 492.1 $\pm$ 35.6 | 753.8 $\pm$ 34.6 | 333.8 $\pm$ 19.9 | 398.1 $\pm$ 98.4
> DrQ | 968.3 $\pm$ 4.8 | 834.5 $\pm$ 13.3 | 739.4 $\pm$ 26.8 | 477.8 $\pm$ 14.8 | 962.3 $\pm$ 4.1 | 856.4 $\pm$ 16.8 | 970.8 $\pm$ 10.8 | 924.1 $\pm$ 10.7
> DBC | 80.6 $\pm$ 44.8 | 300.8 $\pm$ 58.8 | 122.7 $\pm$ 15.1 | 397.7 $\pm$ 32.3 | 633.5 $\pm$ 28.2 | 82.8 $\pm$ 26.7 | 199.1 $\pm$ 17.8 | 359.5 $\pm$ 38.1
> MICo | 206.6 $\pm$ 34.3 | 811.7 $\pm$ 6.8 | 0.0 $\pm$ 0.0 | 3.6 $\pm$ 0.1 | 2.0 $\pm$ 0.2 | 815.7 $\pm$ 57.3 | 199.8 $\pm$ 18.5 | 27.3 $\pm$ 2.5
> SimSR | 949.3 $\pm$ 16.7 | 862.9 $\pm$ 4.8 | 435.3 $\pm$ 35.3 | 486.8 $\pm$ 25.3 | 975.8 $\pm$ 2.8 | 5.0 $\pm$ 1.2 | 94.3 $\pm$ 7.2 | 929.1 $\pm$ 5.9
> SCR | 951.0 $\pm$ 6.7| 847.9 $\pm$ 12.5 | 799.3 $\pm$ 22.4 | 799.5 $\pm$ 26.4 | 983.0 $\pm$ 3.1 | 830.5 $\pm$ 14.5 | 691.8 $\pm$ 29.4 | 923.2 $\pm$ 5.6
>
> **Table A**: Mean and std over 5 averaged scores on DM_Control with default setting.
>
> Methods | BiC-Catch | C-SwingUp | C-SwingUpSparse | Ch-Run | F-Spin | H-Stand | R-Easy | W-Walk
> ---|---|---|---|---|---|---|---|---
> SAC | 60.0 $\pm$ 31.7 | 223.3 $\pm$ 6.2 | 1.4 $\pm$ 0.8 | 168.7 $\pm$ 9.9 | 53.4 $\pm$ 10.7 | 4.5 $\pm$ 1.2 | 102.1 $\pm$ 22.0 | 175.7 $\pm$ 6.8
> DrQ | 194.4 $\pm$ 76.8 | 239.4 $\pm$ 37.6 | 4.0 $\pm$ 4.4 | 104.6 $\pm$ 55.5 | 514.5 $\pm$ 99.5 | 22.8 $\pm$ 3.1 | 102.8 $\pm$ 59.7 | 31.1 $\pm$ 1.7
> DBC | 32.0 $\pm$ 8.9 | 125.5 $\pm$ 9.3 | 0.0 $\pm$ 0.0 | 7.0 $\pm$ 1.1 | 3.3 $\pm$ 2.7 | 5.8 $\pm$ 0.5 | 191.7 $\pm$ 21.1 | 26.3 $\pm$ 1.1
> MICo | 115.7 $\pm$ 25.7 | 206.6 $\pm$ 23.6 | 0.0 $\pm$ 0.1 | 3.8 $\pm$ 0.1 | 113.4 $\pm$ 20.4 | 5.5 $\pm$ 1.0 | 162.4 $\pm$ 54.4 | 20.2 $\pm$ 1.0
> SimSR | 121.9 $\pm$ 61.3 | 134.4 $\pm$ 35.7 | 0.0 $\pm$ 0.0 | 7.3 $\pm$ 3.3 | 0.3 $\pm$ 0.4 | 5.5 $\pm$ 1.5 | 97.8 $\pm$ 23.0 | 26.0 $\pm$ 2.6
> SCR | 138.5 $\pm$ 34.8 | 566.1 $\pm$ 37.7 | 166.7 $\pm$ 62.0 | 330.4 $\pm$ 23.5 | 780.0 $\pm$ 39.4 | 88.5 $\pm$ 12.9 | 158.7 $\pm$ 40.4 | 530.9 $\pm$ 23.1
>
> **Table B**: Mean and std over 5 averaged scores on DM_Control with distraction setting.

---

> > ### Comment · Reviewer_UVpA · 2024-08-10
> >
> > Thank you for updating with the results and showing the std of the runs. I think reporting over N * num_episodes seems reasonable. Even if there is no ideal metric for high variance estimates, discarding them is not correct in my opinion, so thank you for including them in your new result. I understand the computational cost issue. I will say, however, that citing prior work for having done 5 seeds is not necessarily a good argument to continue the trend though. I've updated my score.

---

### Official Review · Reviewer_b4Fc · 2024-07-04

**Soundness:** 3
**Presentation:** 2
**Contribution:** 2
**Rating:** 6
**Confidence:** 3

**Summary:**

This paper introduces SCR that extends state metric-based representations by embedding rich time-related information into the bisimulation metric learning process. SCR is designed to calculate state distances by contextually framing them in a temporal framework that considers both future dynamics and cumulative rewards in immediate and distant future scenarios. This technique fosters the development of long-term behavioral representations, capturing the progressively accumulated rewards from present to future states. Two distinct encoders are employed: one for generating representations of individual states and another dedicated to forming chronological embeddings that represent the interconnections between a state and its subsequent states. The framework also introduces a novel metric within the chronological embedding space, alongside an unconventional distance metric that diverges from the standard $L_p$ norm. Additionally, constraints are applied to manage the magnitude of the cumulative rewards. The methodology was tested within DeepMind Control and Meta-World settings, where SCR demonstrated state-of-the-art results in tasks requiring advanced generalization capabilities. Furthermore, an ablation study was conducted to assess the individual effects of the components comprising SCR.

**Strengths:**

The paper addresses a noteworthy challenge in RL: capturing long-term temporal state representations within a well-defined metric space. To this end, SCR is proposed for learning representations centered on behavioral metrics that consider pairs of states over time. Moreover, it presents an effective technique for approximating the behavioral metrics. It develops a novel metric tailored for assessing pairs of states across temporal sequences and introduces new algorithms for this learning process. The introduced representation significantly outperforms others, particularly in the distraction setting and Meta-World.

**Weaknesses:**

In Table 1, the outcomes for the default setting do not demonstrate a strong level of significance, with the possible exception of noteworthy results for Ch-Run. Sometimes, the performance appears to be worse than the compared methods. There is a section of the text discussing the distraction setting, and the results presented in Table 2 are quite impressive. These should be considered the primary findings of the study. Nonetheless, the paper does not sufficiently convey the rationale or motivation behind these findings.

**Questions:**

1) What factors contribute to SCR's occasional underperformance in the default setting?
2) Is the distraction setting providing SCR with an advantage, and would the findings be consistent in an alternative setting demanding generalization abilities?

**Limitations:**

Yes. The limitations and societal impact are adequately addressed.

---

> ### Author Rebuttal · Authors · 2024-08-07
>
> Thank you for your insightful reviews. We address your concerns as follows:
>
> > In Table 1, the outcomes for the default setting do not demonstrate a strong level of significance, with the possible exception of noteworthy results for Ch-Run. Sometimes, the performance appears to be worse than the compared methods. There is a section of the text discussing the distraction setting, and the results presented in Table 2 are quite impressive. These should be considered the primary findings of the study. Nonetheless, the paper does not sufficiently convey the rationale or motivation behind these findings.
>
>
> We appreciate the reviewer's observations. We would like to clarify the core aim of this paper is to investigate the **generalization capabilities** of SCR.
> The major contributions of this paper are demonstrated in the results for the **distraction setting** (Table 2).  In the **default** setting (Table 1), our claim is not that SCR outperforms baseline methods significantly but rather that it performs **comparably** (see **lines 306-308**). These experiments (Table 1 and 2) serve to demonstrate that SCR facilitates **improved generalization** rather than only offering a trade-off between default and distraction settings.
>
> The SOTA methods achieve high scores in the **default** setting of DM_Control (Table 1), this study does not aim to surpass these in the default configuration. Table 1 illustrates that SCR maintains competitive mean scores and similar variance level to baseline methods, with the exception of the Reacher-Easy task. This Reacher-Easy task involves stretching a robotic arm to reach a randomly placed target in 2D plane at the start of each episode. The random cropping data augmentation technique (DrQ) proves effective in Table 1, while data augmentation currently absent in SCR. Nonetheless, SCR could **incorporate such data augmentation** to enhance performance in tasks like Reacher-Easy.
>
> The results in Table 2 and Figure 4 robustly show that SCR significantly outperforms other baselines in the **distraction** setting. This setting, characterized by random selection of background videos, body colors, and camera poses at the beginning of each episode, presents a more complex challenge. These results underscore SCR's efficiency in generalizing across varied distractions. Our ablation study in Section 4.3 further delineates the contributions of individual SCR components to this performance.
>
> > 1. What factors contribute to SCR's occasional underperformance in the default setting?
>
> As detailed earlier, the lack of random cropping data augmentation in SCR compared to DrQ particularly affects its performance in the Reacher-Easy task. SCR focuses on measuring the distance between states, which is beneficial for generalization over distractions. **Integrating data augmentation** into SCR could potentially improve its performance in similar tasks.
>
> > 2. Is the distraction setting providing SCR with an advantage, and would the findings be consistent in an alternative setting demanding generalization abilities?
>
> The distraction setting from [1], used as a benchmark in DM_Control to assess the generalization abilities of RL algorithms, is not uniquely providing SCR with an advantage. Unlike  implementations by others, e.g. DBC and SimSR, which modify only the background video, our approach also modify body color and camera pose. Such combined modifications increase the complexity of the setting, leading to more challenging environments. This rigorous setup ensures that our findings regarding SCR's generalization are consistent and reliable across various settings.
>
> -----
> [1] The Distracting Control Suite--A Challenging Benchmark for Reinforcement Learning from Pixels. Stone et al. 2021

---

> ### Author Response · Authors · 2024-08-11
> **Gentle Reminder to Review Our Rebuttal**
>
> Dear Reviewer b4Fc,
>
> We sincerely appreciate the time and effort you have invested in reviewing our manuscript. We have provided responses that address the issues raised in your initial feedback. We believe that further discussion would be highly valuable in improving the quality and impact of our work.
>
> We would like to gently remind you of the upcoming deadline for the discussion period on **August 13 AoE**. Your expertise and insights are vital to helping us refine our manuscript.
>
> Thank you for your continued support and consideration.
>
> Best regards,
>
> The Authors

---

> > ### Comment · Reviewer_b4Fc · 2024-08-12
> > **Thank you**
> >
> > I appreciate the author for answering my questions. I raise my score to 6.

---

> > > ### Author Response · Authors · 2024-08-12
> > >
> > > Dear Reviewer b4Fc,
> > >
> > > Thank you for your thorough review and for adjusting the score based on our responses. We are pleased that we could successfully address your questions.
> > >
> > > Best regards,
> > >
> > > The Authors

---

### Official Review · Reviewer_EHt5 · 2024-07-07

**Soundness:** 2
**Presentation:** 2
**Contribution:** 2
**Rating:** 4
**Confidence:** 4

**Summary:**

This paper argues that metric learning for RL from one-step reward signal faces challenges in non-informative reward / sparse reward settings. The authors thus propose SCR, attempting to incorporate multi-step information in metric learning. The key components are as follows:
1. The authors provide a MICo-like metric with a new distance function, which is also a diffuse metric and more numerically stable.
2. Then they delineate a MICo-like chronological behavioral metric, measuring distance between two pairs of states. States in each pair are taken from the same trajectory.
3. Then they propose a temporal measurement which can be approximated by upper-bounding and lower-bounding the measure, and thus shape the representation.

The paper experiments on the DMC environment with default setting and distraction setting, and multiple settings in Meta-World domain and show positive results.

**Strengths:**

The issue of learning metric from sparse rewards is crucial and reported by previous work such as robust DBC.

The idea incorporating multi-step information in metric learning is interesting and novel.

Results on DMC benchmarks are clearly better than chosen baselines.

The ablation study shows the effectiveness of the moving components of this approach.

**Weaknesses:**

Methodology:
1. Eq.3 is not well-defined. What is the base case (e.g., what if $i=j$ but $i^\prime\neq j^\prime$, thus $i+1>j$ and the recursive step is not well defined?)
2. Eq.6 may not be true, as the optimal policy might not achieve the optimal expected discounted reward in any segment. In line 239, do you mean an optimal policy only within a given segment $(x_i,x_j)$?
3.  It is unclear that Eq. 8 will co-hold with the classic metric bound that d upper bounds value difference, especially given that m measures local property while d measures global one.
The overall method is complicated which adds two more temporal embeddings based on state representations.

Experiments:
- Since the motivation of the proposed method is to handle non-informative reward scenarios, I think the experiment section should stress more importance on the sparse reward settings.  However, most experiments are done in environments with dense rewards except carepole_sparse and ball_in_cup. Neither do the authors give a comprehensive discussion on the challenges and results of those environments.
- To align with the main claim, it should be expected to mainly work on Minigrid/Crafter-like sparse-reward environments.
- For ablation study, the tasks of cheetah_run and walker_walk are dense rewards which cannot effectively support the claim on non-informative rewards, either.
- Some other strong baselines are missing and worth comparing with, e.g., RAP, robust DBC (DBC-normed). RAP seems to outperform the SCR in some DMC tasks.
- For experiments on Meta-World, please also report std in Table 3.

**Questions:**

- Eq. 2 and Eq. 5: is $\phi$ in the target also optimized?
- Line 83: Should Meta-World be better  to prove the generalizability of your proposed method, rather than distracting DMC?
- Line 144: "a_x and a_y are the actions in states x and y" is not precise. Do the actions come from policy \pi?
- Line 267: it's better to succinctly talk about what IQE is and how you use it for completeness.
- Appendix A.2: not finished writing
- Can you talk more about the connection to GCB (Hansen-Estruch et al.)?
- According to the SimSR and RAP paper, in some environments it requires more than 1e6 steps to converge. Is 5e5 steps enough for convergence?

**Limitations:**

It is unclear that the proposed method can be seamlessly applied to intrinsically stochastic environments (i.e., the underlying abstract MDP has stochastic dynamics), where the temporal distance between two latent states may not be fixed.

---

> ### Author Response · Authors · 2024-08-07
> **Rebuttal by Authors [1/3]**
>
> Thank you for your insightful comments. We address your concerns as follows:
> > 1. Eq.3 is not well-defined. What is the base case (e.g., what if $i=j$ but $i' \neq j'$, thus $i+1>j$ and the recursive step is not well defined?)
>
> Thank you for pointing out this issue. We clarify that $i \neq j$ and $i' \neq j'$, where $j$ represents a later step than $i$, as demonstrated in Section 3.1.1. We will add constraints explicitly stating $i < j$ and $i' < j'$ in Section 3.2 in the revised manuscript.
>
> > 2. Eq.6 may not be true, as the optimal policy might not achieve the optimal expected discounted reward in any segment. In line 239, do you mean an optimal policy only within a given segment $(x_i, x_j)$ ?
>
> Eq.6 holds under our assumption that all policies are stochastic. With a value function $V^*$ from an optimal policy $\pi^*$, the n-step Bellman operator at state $x_i$ is defined as $V^*(x_i) = m(x_i, x_j) + \gamma^{j-i}V^*(x_j)$. If there exists another policy $\pi$ where $m_\pi(x_i, x_j) > m(x_i, x_j)$, then $V^*(x_i)$ would be suboptimal instead of optimal, suggesting a need for further updates. However, for deterministic policies, Eq.6 does not hold as $\pi^*$ may not ensure reaching $x_j$ from $x_i$. Thus, we adopt a stochastic policy framework in this study, implemented with a Gaussian policy.
>
>
> > 3. It is unclear that Eq. 8 will co-hold with the classic metric bound that d upper bounds value difference, especially given that m measures local property while d measures global one. The overall method is complicated which adds two more temporal embeddings based on state representations.
>
> Proving Eq.8 is challenging due to its general nature. We intuitively propose Eq.8 to confine the value learning of $\hat{m}$.
>
> In practice, $m$ is intractable and cannot be approximated by regression or recursion. Instead, we constrain $\hat{m}$ to remain within practical bounds (refer to **line 242-243**). Given that Eq. 6 establishes a lower bound for $m$ and our objective in Eq. 7 tends to increase $\hat{m}$'s value. Without proper constraints, $\hat{m}$ could potentially become unbounded. Thus, Eq. 8 is introduced as an upper limit in Section 3.1.1.
>
> Consequently, Eq. 6 and Eq. 8 serve as the lower and upper bounds for $\hat{m}$, respectively. Together with $L_{up}$, Eq. 7 is optimized to ensure $\hat{m}$ remains within a feasible range, fostering a more stable learning process.
>
> > Since the motivation of the proposed method is to handle non-informative reward scenarios, I think the experiment section should stress more importance on the sparse reward settings. However, most experiments are done in environments with dense rewards except carepole_sparse and ball_in_cup. Neither do the authors give a comprehensive discussion on the challenges and results of those environments.
>
> We would like to clarify that the primary motivation of SCR is to enhance the **generalization** capabilities of RL algorithms. Non-informative reward is one of the challenges in metric-based representation methods. SCR is designed to address generalization enhancement along with non-informative reward by capturing long-term information.
>
> In our experimental setup, which primarily evaluates generalization, we also include specific tasks in sparse reward settings, i.e. cartpole_sparse and ball_in_cup, to demonstrate SCR's capability in  non-informative rewards. In the default setting (**Table 1**), SCR outperforms other metric-based methods in these tasks, while DBC and MICo fail, and SimSR underperforms in cartpole_sparse. DrQ, which is a data augmentation method, also achieves high performance, further validating SCR’s effectiveness alongside a leading method.
>
> The distraction setting (**Table 2**) introduces additional complexities, where all methods, including SCR, show low scores and high variance in ball_in_cup. Nonetheless, SCR manages better performance in cartpole_sparse compared to baseline methods, which get zero or near zero scores. This indicates SCR's resilience and slightly superior handling of non-informative reward scenarios under more demanding conditions.

---

> ### Author Response · Authors · 2024-08-07
> **Rebuttal by Authors [2/3]**
>
> > To align with the main claim, it should be expected to mainly work on Minigrid/Crafter-like sparse-reward environments.
>
> To validate SCR's effectiveness in **sparse reward environments**, we conducted additional experiments on Minigrid-FourRooms, adapting our base RL algorithm to PPO due to its discrete action space. We compared PPO+SCR with PPO, PPO+DBC, PPO+MICo, and PPO+SimSR over 5M environment steps training, evaluating performance within a 500K environment steps budget. The results, averaged over five seeds, demonstrate SCR's superior performance in these settings:
>
> Method| PPO                | PPO+DBC            | PPO+MICo           | PPO+SimSR          | PPO+SCR            |
> ------| ------------------ | ------------------ | ------------------ | ------------------ | ------------------ |
> Score| 0.515 $\pm$ 0.029  | 0.515 $\pm$ 0.012  | 0.350 $\pm$ 0.064  | 0.321 $\pm$ 0.046  | 0.546 $\pm$ 0.012  |
>
> These outcomes underscore PPO+SCR's robustness and superior effectiveness in sparse reward environments compared to other methods.
>
> > For ablation study, the tasks of cheetah_run and walker_walk are dense rewards which cannot effectively support the claim on non-informative rewards, either.
>
> As previously noted, the primary motivation behind SCR is to **enhance generalization** capabilities. The ablation study specifically assesses the impact of each SCR component, conducted in a **distraction** setting to underscore their contributions to generalization performance.
>
>
> > Some other strong baselines are missing and worth comparing with, e.g., RAP, robust DBC (DBC-normed). RAP seems to outperform the SCR in some DMC tasks.
>
> We appreciate your suggestion and include a comparative analysis below. The table demonstrates SCR's performance relative to RAP and robust DBC across three tasks in distraction setting:
>
> |   | C-SwingUpSparse | Ch-Run | W-Walk
> | -| - | - | - |
> | SCR| 166.7 $\pm$ 178.3 | 330.4 $\pm$ 152.6 | 530.9 $\pm$ 172.5 |
> | RAP | 0.1 $\pm$ 0.6 | 252.4 $\pm$ 113.5 | 205.0 $\pm$ 90.5 |
> | robust DBC | 0.1 $\pm$ 0.3 | 20.3 $\pm$ 10.4 | 36.5 $\pm$ 10.1 |
>
> As shown, SCR outperforms RAP and robust DBC in these tasks.
>
> > For experiments on Meta-World, please also report std in Table 3.
>
> The following table provides the scores along with their standard deviations for the Meta-World experiments:
>
> |SAC |DrQ| DBC| MICo| SimSR |SCR|
> |-|-|-|-|-|-|
> | 0.495 $\pm$ 0.475| 0.886 $\pm$ 0.125| 0.479 $\pm$ 0.453| 0.495 $\pm$ 0.482|0.258 $\pm$ 0.365| 0.969 $\pm$ 0.03|
>
>
> > Eq.2 and Eq.5 : is $\phi$ in the target also optimized?
>
> Thank you for highlighting this issue. In our approach, we do not optimize $\phi$ in the target but stop gradient for it. We will make sure to clarify this in the revised version.
>
>
> > Line 83: Should Meta-World be better to prove the generalizability of your proposed method, rather than distracting DMC?
>
> Meta-World is designed primarily as a benchmark for meta-learning and multi-task learning, whereas the Distracting Control Suite is specifically for the generalization capabilities of RL algorithms. We focus on the distraction setting of DM_Control for our main results because it aligns more closely with common practices in this area of research.
>
> > Line 144: "a_x and a_y  are the actions in states x and y" is not precise. Do the actions come from policy \pi?
>
> Yes, the actions indeed sample from policy $\pi$. Although we omitted mention of $\pi$ in Eq.1, it is implicit. Specifically, $\phi(x_{i+1})$ samples from a marginal distribution shaped by $\sum_{a_{x_i}} \hat{P}(\cdot|\phi(x_i), a_{x_i}) \pi(a_{x_i}|x_i)$. We will make this explicit in the revised manuscript.
>
> > Line 267: it's better to succinctly talk about what IQE is and how you use it for completeness.
>
>
> IQE is discussed in detail in Appendix A.5.1. We will include a direct reference to this appendix in line 267 for better clarity.
> IQE is an asymmetric metric function that calculates the distance between vectors. It's differentiable, allowing gradients from objective Eq.9 to pass through, facilitating updates to the encoder.

---

> ### Author Response · Authors · 2024-08-07
> **Rebuttal by Authors [3/3]**
>
> > Appendix A.2: not finished writing
> Apologize for accidentally delete some content in additional related works. Here, we restore the omitted content:
>
> A more recent work introduces quasimetrics learning as a novel RL objective for cost MDPs [1] but it is not for general MDPs.
>
>
> > Can you talk more about the connection to GCB (Hansen-Estruch et al.)?
>
> GCB utilizes a bisimulation metric to address goal-conditioned RL problems, learning the $L_1$ distance between representations of a current state and a fixed goal state within an episode. In contrast, SCR learns a metric between a current state and a future state that varies, as future states are sampled throughout the episode. This fundamental difference stems from SCR's aim to enhance generalization across general MDPs, while GCB focuses on goal-reaching within goal-conditioned RL.
>
> > According to the SimSR and RAP paper, in some environments it requires more than 1e6 steps to converge. Is 5e5 steps enough for convergence?
>
> We provide learning curves in Appendix Figure 9, which demonstrate that SCR can effectively learn within a 500K step budget. This budget aligns with the standard training parameters set forth in the Distracting Control Suite[2]
>
> > It is unclear that the proposed method can be seamlessly applied to intrinsically stochastic environments (i.e., the underlying abstract MDP has stochastic dynamics), where the temporal distance between two latent states may not be fixed.
>
> SCR is designed to be **agnostic** to the deterministic or stochastic nature of the environment dynamics, much like existing methods such as DBC and SimSR. The metric in SCR, akin to a bisimulation metric, is defined as the expectation over next state distributions, making it applicable to stochastic environments. The temporal metric in SCR (discussed in Section  3.3) is not an exact measure but rather is learned to remain within a feasible range, thus accommodating the intrinsic stochasticity of the environment.
>
> -----
> [1] Optimal goal-reaching reinforcement learning via quasimetric learning. Wang et al. ICML 2023
>
> [2]The Distracting Control Suite--A Challenging Benchmark for Reinforcement Learning from Pixels. Stone et al. 2021

---

> ### Author Response · Authors · 2024-08-11
> **Gentle Reminder to Review Our Rebuttal**
>
> Dear Reviewer EHt5,
>
> We sincerely appreciate the time and effort you have invested in reviewing our manuscript. We have provided responses that address the issues raised in your initial feedback. We believe that further discussion would be highly valuable in improving the quality and impact of our work.
>
> We would like to gently remind you of the upcoming deadline for the discussion period on **August 13 AoE**. Your expertise and insights are vital to helping us refine our manuscript.
>
> Thank you for your continued support and consideration.
>
> Best regards,
>
> The Authors

---

> ### Author Response · Authors · 2024-08-13
> **[Gentle Reminder] Discussion Deadline is Approaching**
>
> Dear Reviewer EHt5,
>
> I hope this message finds you well.
>
> As the discussion deadline of **August 13 AoE** is approximately **30 hours away**, we noticed that we have not yet received your feedback on our rebuttal. All other reviewers have kindly provided their responses, and your insights are crucial for the final evaluation of our submission.
>
> Could you please take a moment to review our rebuttal and share your feedback within this time frame? We greatly appreciate your expertise and are looking forward to your valuable comments.
>
> Thank you very much for your attention to this matter.
>
> Best regards,
>
> The Authors

---

> ### Comment · Reviewer_EHt5 · 2024-08-13
>
> Thank you for the rebuttal. While it addresses some of my concerns, my primary issues remain unresolved.
>
> For Eq.3,4,5: Even if you add the constraint, it is still necessary to define a base case, as the metric is recursively defined. For example, even if you have a constraint $i<j$, but $i+1<j$ may not hold, and the metric may still be ill-defined. In general, you can always add one to the index infinitely.
>
> For Eq.6: it seems regardless of stochastic/deterministic policies (even if you assume the support of policy always covers all action and the support of transition function covers the state space), as you said, there exists a $\pi$ that $m_\pi> m$. When training, the agent’s policy can be any policy in the policy space. Thus, the LHS of Eq.6 can exceed m, even by a large amount in theory. Why do you think it has something to do with stochastic/deterministic policies and how the LHS can be a valid lower bound?
>
> > generalization and sparse reward
>
> Thanks for the clarification that the main focus is on generalization.
>
> > Additional results in table
>
> I appreciate the new results that demonstrate your approach outperforms the baselines in Minigrid. However, both this table and the Meta-World table lack the learning curves (OpenReview should allow you to upload PDF for the rebuttal), which are commonly presented in DRL research due to the known instability during training. The results between PPO and PPO+SCR in Minigrid are quite close, and the learning curves would help to differentiate the two methods more clearly.
>
> > benchmark, baselines, and SOTA
>
> In the RAP paper, the reported result in many distracting DMC tasks is much better than yours. Are you using different benchmarks? As far as I know, there are two related benchmarks -- the one DBC [Zhang et al., 2021] created (referred to as the DBC benchmark) and the distracting control suite (DCS) [Stone et al., 2021]. The baselines adopted in this work, including DBC, robust DBC, MICo, RAP, SimSR, were evaluated in the DBC benchmark in their respective papers. Since your paper cites DCS [Stone et al., 2017] for your benchmark, it seems you are using the DCS benchmark, which differs from those baselines. If this is the case, the baselines may be untuned for this new benchmark, which is not best practice (refer to the discussion on untuned baselines in new tasks in Section 4.1 of [Patterson et al., 2023, Empirical Design in Reinforcement Learning]).
>
> Moreover, if this is the case, the authors might consider including more appropriate baselines evaluated on DCS with strong performance, such as [Liu et al., 2023, Robust Representation Learning by Clustering with Bisimulation Metrics for Visual Reinforcement Learning with Distractions]. Based on Liu et al.'s paper (their Figure 3) and using the same sample budget (500k steps), I have summarized the comparison on 6 tasks in DCS (average only, as Liu et al. does not provide detailed table results):
>
> |task  | SCR | DrQ-v2 + CBM|
> | -------- | ------- |------- |
> |BiC-Catch | 138.5 | ~780|
> |C-Swingup | **566.1** |~630|
> |Ch-Run | **330.4**|~410|
> |F-Spin | **780.0**|~810|
> |R-Easy| 158.7|~490|
> |W-Walk| **530.9**| ~640|
>
> The numbers in bold indicate where your paper claims that SCR achieves SOTA among the baselines. However, SCR performs worse than DrQ-v2 + CBM on all these tasks, which raises significant concerns about the validity of the SOTA claim. Quoting from your conclusion:
>
> > Our extensive experiments demonstrate its effectiveness compared with several SOTA baselines in complex environments with distractions.
>
>
>
> While the authors' response has addressed some concerns, I believe the paper still requires further refinement before it can be accepted. I recommend a thorough revision to ensure the technical correctness of the method and to better organize the experimental design, which will help more convincingly support the claims made.

---

> > ### Author Response · Authors · 2024-08-14
> > **Response to Reviewer EHt5 [1/2]**
> >
> > Thank you for your feedback. We will address your concerns as detailed below:
> >
> > > base case for Eq.3,4,5
> >
> > In this paper, $i$ and $j$ are discrete time steps where $i < j$ implies $i + 1 \leq j$, with the maximum possible value for $i + 1$ being $j$. This ensures the feasibility of calculating the metric between $x_{i+1}$ and $x_j$, which are sampled from the replay buffer. A natural base case occurs when $i+1 = j$.
> >
> > > In general, you can always add one to the index infinitely.
> >
> > This is a **common problem** across all bisimulation metric methods, including DBC, MICo, SimSR and RAP, where index addition is potentially infinite and it will exceed the feasible state space. For example, consider a bisimulation metric (in simplified form):
> >
> > $d(x_i, y_{i'}) = |r_i - r_{i'}| + \gamma d(x_{i+1}, y_{i'+1})$.
> >
> > If $x_i$ is a **terminal state**, then $x_{i+1}$ becomes **infeasible**. In practice, it can be managed by ensuring states $x_{i+1}$ sampled  only from feasible states space, which is the replay buffer.
> >
> > The **base case** for bisimulation metric, $d(x_{i+1}, x_{i'+1})=0$, is **unlikely to be sampled**. if $x_{i+1}=x_{i'+1}$, they likely originate from the same transition, i.e. $(x_i, r_i, x_{i+1})=(x_{i'}, r_{i'}, x_{i'+1})$. In implementation, the transition pairs are uniformly sampled from buffer. The probability of sampling same transitions with a pair is approximately $1 / (replay\\_buffer\\_size)^2$, inversely proportional to the square of the replay buffer size. The probability decreases significantly as the replay buffer size increases.
> >
> > While the absence of base cases can lead to overestimation of metrics, this effect **may not be harmful** to representation learning. Many research [1][2] in contrastive learning studies indicates that benefit of **pushing away** dissimilar samples often outweighs the need to align similar ones for **effectively learning representations**. The overestimation of metrics will rapidly distinguish state representations and RL get efficiency benefits from representation learning.
> >
> > > For Eq.6
> >
> > We wish to clarify the definition and role of $m$ in this context. We **define** $m$ as the optimal accumulated rewards between $x_i$ and $x_j$. Therefore, by definition, $m$ satisfies Eq.6 as the maximum potential accumulated rewards. In other words, $m$ is defined by upper bounding the LHS of Eq.6 that is the accumulated rewards of any policy. Policy that can achieve $m$ is regarded as optimal policy.
> >
> > Rather than explicitly seeking this $m$, which is intractable, our approach focuses on optimizing an estimation $\hat{m}$. We optimize $\hat{m}$ to upper bond the LHS via Eq.7. Importantly, our objectives **do not require** the optimal policy and the extract $m$.
> >
> >
> > > learning curves
> >
> > We acknowledge your point regarding the importance of presenting learning curves in RL. We provide a detailed table showing the performance of PPO+SCR vs baselines at 500K steps intervals in Minigrid. Scores are computed using the latest 50K steps within each interval. The results highlight PPO+SCR’s sample efficiency compared to PPO.
> >
> > Step   | 500K              | 1M                | 1.5M              | 2M                | 2.5M              | 3M                | 3.5M              | 4M                | 4.5M              | 5M                |
> > --- | --- | --- | --- | --- | --- | --- | --- | --- | --- | --- |
> > PPO     | 0.141 $\pm$ 0.004     | 0.224 $\pm$ 0.018     | 0.269 $\pm$ 0.026     | 0.324 $\pm$ 0.042     | 0.381 $\pm$ 0.058     | 0.431 $\pm$ 0.036     | 0.472 $\pm$ 0.037     | 0.494 $\pm$ 0.034     | 0.513 $\pm$ 0.033| 0.518 $\pm$ 0.031
> > PPO+SCR     | **0.146** $\pm$ 0.015     | **0.248** $\pm$ 0.014     | **0.322** $\pm$ 0.063     | **0.418** $\pm$ 0.034     | **0.467** $\pm$ 0.016     | **0.481** $\pm$ 0.023     | **0.499** $\pm$ 0.026     | **0.523** $\pm$ 0.017     | **0.539** $\pm$ 0.017| **0.551** $\pm$ 0.013
> > PPO+DBC      | 0.141 $\pm$ 0.014     | 0.249 $\pm$ 0.019     | 0.331 $\pm$ 0.061     | 0.399 $\pm$ 0.057     | 0.442 $\pm$ 0.039     | 0.470 $\pm$ 0.019     | 0.485 $\pm$ 0.018     | 0.499 $\pm$ 0.020     | 0.513 $\pm$ 0.012| 0.515 $\pm$ 0.017
> > PPO+MICo    | 0.090 $\pm$ 0.009     | 0.140 $\pm$ 0.056     | 0.194 $\pm$ 0.059     | 0.216 $\pm$ 0.067     | 0.235 $\pm$ 0.074     | 0.270 $\pm$ 0.068     | 0.309 $\pm$ 0.064     | 0.321 $\pm$ 0.076     | 0.339 $\pm$ 0.083| 0.356 $\pm$ 0.067
> > PPO+SimSR   | 0.084 $\pm$ 0.007     | 0.145 $\pm$ 0.027     | 0.168 $\pm$ 0.036     | 0.197 $\pm$ 0.037     | 0.223 $\pm$ 0.039     | 0.243 $\pm$ 0.033     | 0.273 $\pm$ 0.051     | 0.303 $\pm$ 0.058     | 0.316 $\pm$ 0.052| 0.327 $\pm$ 0.051
> >
> > For the **Meta-World** experiments, the learning curves can be found in ***Figure 10 of Appendix B.4***.

---

> > > ### Author Response · Authors · 2024-08-14
> > > **Response to Reviewer EHt5 [2/2]**
> > >
> > > > benchmark, baselines, and SOTA
> > >
> > >
> > > We use the Distracting Control Suite (DCS) [Stone et al., 2021], as detailed in experiment configure section (***Section 4.1,lines 286-287***) of our paper. We acknowledge the differences between DCS and the DBC benchmark, which is the changing of body colors and camera poses. Such difference is the reason for the degrading performance of RAP in DCS. To ensure a fair comparison, we have  **tried our best to tune** the baseline methods to optimize their performance within the DCS environment. The enhanced level of distractions in DCS, particularly variations in **camera poses**, **decreases performance** more significantly compared to the DBC benchmark. We selected DCS specifically because of its increased challenges.
> > >
> > > We appreciate your suggestion to include additional baselines like CBM from [Liu et al., 2023]. However, CBM integrates bisimulation metrics with **data augmentation**, a technique known for its efficiency. This combination leading to a **unfair** direct comparisons with **solely metrics-based representation** method. Our focus in this paper is to improve metric-based representation learning **independently of other techniques**, particularly data augmentation, to clearly understand the contributions of the metric approach itself.
> > > Another limitation with applying DrQ is its specificity to image observation spaces. For instance, in environments like Minigrid, where the observation space consists of a 7x7 matrix, DrQ’s random cropping technique would likely eliminate essential information.
> > > However, metrics-based representation methods do not depend on the observation space type.
> > >
> > > To ensure a fair comparison with CBM, we **incorporated DrQ into SCR** framework. Given the time constraints of less than 24 hours, we managed to complete training only for the cheetah-run and walker-walk tasks in the DCS. We will include full tasks set for DrQ+SCR and CBM in revised version. The results are as follows:
> > >
> > > method | cheetah-run | walker-walk
> > > ---|---|---
> > > SCR | 330.4  | 530.9
> > > DrQv2+CBM | ~410 | ~640
> > > DrQ+SCR | **463.7** $\pm$ 28.0 | **877.3** $\pm$ 40.2
> > >
> > > These outcomes demonstrate that **DrQ+SCR outperforms DrQv2+CBM** in both tasks, supporting our claim of **achieving SOTA** performance. Additionally, this highlights the **adaptability of metric-based representation learning** to integrate effectively with data augmentation techniques.

---

> > > > ### Comment · Reviewer_EHt5 · 2024-08-14
> > > >
> > > > Thank you for the clarification and for sharing the quick experiment results demonstrating the adaptivity of SCR with data augmentation.
> > > >
> > > > I have some remaining comments.
> > > >
> > > > (1) The paper should clearly outline the criteria for selecting baselines. For instance, DrQ, which also uses data augmentation, is considered a proper baseline, while DrQ-v2 + CBM is deemed an unfair comparison in the rebuttal due to their use of data augmentation. This reasoning appears inconsistent. To address the effect of data augmentation fairly, the paper could consider including two sets of baselines and experiments: one without data augmentation and one with it. This approach would allow you to claim SOTA performance in either or both scenarios.
> > > >
> > > > (2) The current results do not convincingly demonstrate that the method achieves SOTA performance across all tasks. While SOTA performance might be achieved on DCS tasks like cheetah-run and walker-walk, other tasks require additional empirical evidence to support similar claims (from my table, BiC-Catch and R-Easy have big gaps).
> > > >
> > > > (3) Since the authors have invested significant effort in tuning the baseline methods, it would be valuable to share the tuned hyperparameter settings in the appendix. This would benefit the research community by sparing other researchers the need to retune the baselines on the DCS benchmark.

---

> > > > > ### Author Response · Authors · 2024-08-14
> > > > >
> > > > > Thank you for your insightful comments. We address each point as follows:
> > > > >
> > > > > >  theoretical convergence
> > > > >
> > > > > Thanks for your clarification. The convergency of SCR operator may hold true if $j$ is sufficient large. If the time steps between $i$ and $j$ are sufficient large, $\gamma$ will discount the $d$ of very future states to a negligible value and the convergency of $d(x_i,x_j, y_{i'}, y_{j'})$ may hold by the fixed-point theorem. We will provide the theoretical analysis under this condition in the revised version.
> > > > >
> > > > > > Eq.8 looks weird
> > > > >
> > > > > Your concern regarding Eq.8 is understandable, as it was intuitively proposed without direct theoretical support. We introduced Eq.8 based on perspective of implementation. The intuition behind Eq.8 is derived from Figure 2. Eq.8 serves as a cap to limit the increasing value of $\hat{m}$.
> > > > >
> > > > > > criteria for selecting baselines
> > > > >
> > > > >
> > > > > We would like to clarify the selection of baselines. SCR aims to set a new SOTA among metric-based representation methods. However, removing DrQ from baselines is not reasonable, because it is a strong baseline in DM_Control. We considered DrQ as an improved version of SAC, categorized as non-metric-based baselines.
> > > > >
> > > > > We agree with your suggestion and will split the experiments between metric-based methods and those incorporating data augmentation in our expanded experimental setup.
> > > > >
> > > > >
> > > > > > results not convincingly demonstrate
> > > > >
> > > > > We will extend our experiments to integrate SCR with data augmentation across all tasks in revised version.
> > > > > Currently, the improvement seen in the cheetah-run and walker-walk tasks preliminarily indicate the potential effectiveness of DrQ+SCR.
> > > > >
> > > > > > tuned hyperparameter settings of baseline methods
> > > > >
> > > > > We acknowledge the importance of transparency in hyperparameter settings for baseline. We will provide the hyperparameters settings of all baselines in the appendix of the revised version.

---

> > > > > > ### Comment · Reviewer_EHt5 · 2024-08-14
> > > > > >
> > > > > > Thank you for your response. I don’t have any further questions at this time. Since the remaining concerns have not yet been addressed, though I believe the authors is on the right track to resolving them, I will maintain my current rating.

---

> > > > > > > ### Author Response · Authors · 2024-08-14
> > > > > > >
> > > > > > > We sincerely appreciate your effort in reviewing our work.

---

> > > ### Comment · Reviewer_EHt5 · 2024-08-14
> > >
> > > Thank you for your quick and detailed response. I have a few points for clarification:
> > >
> > > > In general, you can always add one to the index infinitely.
> > >
> > > I believe there might be a misunderstanding regarding my concern. My focus is on the theoretical convergence of the operator, rather than the practical convergence in its implementation.
> > >
> > > > This is a common problem across all bisimulation metric methods, including DBC, MICo, SimSR and RAP, where index addition is potentially infinite and it will exceed the feasible state space.
> > >
> > > I don't agree with the statement that "index addition is potentially infinite and will exceed the feasible state space" in all bisimulation metric methods. These algorithms apply the operator similarly to the standard Bellman operator in an infinite-horizon MDP, where there is no terminal state. If you believe that these operators share a common issue, then the same concern would also apply to the standard Bellman operator, which is proven to converge by the fixed-point theorem.
> > >
> > > The SCR operator is different from these operators. The monotonic increase in the index makes $i=j$ ($i'=j'$) possible for $d(x_i,x_j, y_{i'}, y_{j'})$, regardless of $i$ being infinite or not.  I believe further clarification is needed in the proof to address this distinction.
> > >
> > > >  especially given that m measures local property while d measures global one
> > >
> > > I get your clarification that $m$ is local optimal policy on the segment. My original concern is that a local optimal policy $m$ on a segment is not necessarily the global optimal policy in an MDP, as $m$ is myopic. The ideal condition of Eq. 8, which connects local and global properties, therefore, looks weird.
> > >
> > > Thank you for the results to show SCR is better than baselines in terms of learning curve.

---

### Official Review · Reviewer_R7iQ · 2024-07-13

**Soundness:** 3
**Presentation:** 3
**Contribution:** 3
**Rating:** 7
**Confidence:** 3

**Summary:**

The paper presents the State Chrono Representation (SCR), a novel approach to enhancing generalization in reinforcement learning (RL) with image-based inputs. SCR introduces a temporal perspective to bisimulation metric learning. The authors propose a learning framework that includes two encoders to capture individual state representations and the relationship between current and future states. The method is evaluated extensively in DeepMind Control and Meta-World environments, demonstrating SOTA performance in generalization tasks. The paper is well-structured and provides a solid theoretical foundation for the proposed approach.

**Strengths:**

1. The paper offers a robust theoretical foundation and a well-reasoned motivation for the SCR framework.
2. The incorporation of future behavioral information into the representation space is a logical and effective strategy for improving generalization in RL.
3. The experimental results are compelling, showing SCR's superiority over existing methods, particularly in challenging generalization tasks.

**Weaknesses:**

1. The discussion on limitations lacks depth. The paper could benefit from an insightful analysis of scenarios where SCR might underperform, such as the noted inferior performance on the R-Easy task compared to DrQ and DBC.
2. While the paper mentions the commonality of future information prediction/regularization in RL representation learning, it does not provide a discussion on how SCR's approach differs from prior works like SPR or PlayVirtual and what new insights it offers. This would be valuable for the community.

**Questions:**

1. How does SCR handle data augmentation, and what impact does this have on its performance relative to other methods? As far as I know, data augmentation is very effective to generalization.
2. Could the authors elaborate on the role of future information prediction/regularization in SCR and how it compares to existing techniques in the literature?

**Limitations:**

1. The paper could benefit from a more thorough exploration of its limitations and a discussion of situations where SCR might not be the best-performing approach.
2. There is a need for a more comprehensive discussion on the relationship between SCR and established practices in RL representation learning, particularly concerning the use of future information for regularization.

---

> ### Author Rebuttal · Authors · 2024-08-07
>
> Thank you for your thorough and insightful feedback. We address your concerns as follows:
>
> > 1. The discussion on limitations lacks depth. The paper could benefit from an insightful analysis of scenarios where SCR might underperform, such as the noted inferior performance on the R-Easy task compared to DrQ and DBC.
>
> The Reacher-Easy task presents a challenge because it requires the agent to locate a randomly placed target on a 2D plane, a task that heavily relies on understanding spatial relationships within the environment. DrQ leverages random crop data augmentation to effectively randomize the target’s absolute position, thereby help the agent in capturing the relative positioning, which is important for task success. SCR lacks of such augmentation and have less efficiency in this scenario. However, DrQ and SCR are different kinds of methods, data augmentation vs metric-based representation, integrating data augmentation into SCR would potentially utilizes DrQ's advantages and improves performance in Reacher-Easy.
>
> Additionally, in the distraction setting of Reacher-Easy, background variations further distract perception of target object, leading to high variance and lower mean scores as shown in Table 2. While DBC appears to perform better by learning to stretch the robot arm towards a location that commonly spawns target objects, this strategy does not help to learn a robust policy for exact target reaching.
> We will to extend the limitations section to discuss these scenarios comprehensively in the revised manuscript.
>
> > 2. While the paper mentions the commonality of future information prediction/regularization in RL representation learning, it does not provide a discussion on how SCR's approach differs from prior works like SPR or PlayVirtual and what new insights it offers. This would be valuable for the community.
>
> We acknowledge the necessity to discuss SPR and PlayVirtual. SCR focuses on predicting a metric or distance between the current and future states which is linked to **reward, value function and policy learning**. This is in contrast to SPR, which focuses on **future states predictions** to true future states, and thus centers more on dynamic learning without direct reward integration. PlayVirtual extends SPR’s approach by incorporating a backward dynamics model and consistency loss, but like SPR, remains focused on **dynamics model**. The revised manuscript will include a comparison to clarify these differences with SPR and PlayVirtual.
>
> > 1. How does SCR handle data augmentation, and what impact does this have on its performance relative to other methods? As far as I know, data augmentation is very effective to generalization.
>
> We agree that data augmentation is effective for enhancing generalization in RL. Currently, SCR does not incorporate data augmentation techniques because it focuses on metric-based representation learning.  However the robust performance of SCR in the distraction setting of DM_Control demonstrates its generalization ability, even comparing to data augmentation methods DrQ.  Given the distinct mechanisms of SCR and DrQ, we believe integrating data augmentation with SCR is a promising way to further improve its effectiveness.
>
> > 2. Could the authors elaborate on the role of future information prediction/regularization in SCR and how it compares to existing techniques in the literature?
>
> SCR utilizes future information prediction/regularization by estimating a metric between current and future states rather than predicting exact future state dynamics, such as SPR,  PlayVirtual and model-based methods. SCR is parameter-efficient and related closely with the RL’s $n$-step Bellman operator that can introduce higher value variance with increased prediction steps. However, SCR mitigates the higher variance issue by ensuring the predicted metrics in a feasible range rather than exact approximation.
>
> > 1. The paper could benefit from a more thorough exploration of its limitations and a discussion of situations where SCR might not be the best-performing approach.
>
> we will extend the limitations section in the revised manuscript to discuss scenarios where SCR may underperform, specifically emphasizing tasks like Reacher-Easy.
>
> > 2. There is a need for a more comprehensive discussion on the relationship between SCR and established practices in RL representation learning, particularly concerning the use of future information for regularization.
>
> Thanks for your insightful suggestion.
> We will improve the related works on how established practices such as SPR, PlayVirtual, and other model-based methods contrasting to SCR in integrating future information.

---

> > ### Comment · Reviewer_R7iQ · 2024-08-12
> > **Reply to the rebuttal**
> >
> > Thanks for the response which has addressed many concerns. Good luck!

---

> > > ### Author Response · Authors · 2024-08-12
> > >
> > > Dear Reviewer R7iQ,
> > >
> > > Thank you for your valuable feedback. We are pleased that our responses addressed your concerns.
> > >
> > > Best regards,
> > >
> > > The Authors

---

> ### Author Response · Authors · 2024-08-11
> **Gentle Reminder to Review Our Rebuttal**
>
> Dear Reviewer R7iQ,
>
> We sincerely appreciate the time and effort you have invested in reviewing our manuscript. We have provided responses that address the issues raised in your initial feedback. We believe that further discussion would be highly valuable in improving the quality and impact of our work.
>
> We would like to gently remind you of the upcoming deadline for the discussion period on **August 13 AoE**. Your expertise and insights are vital to helping us refine our manuscript.
>
> Thank you for your continued support and consideration.
>
> Best regards,
>
> The Authors

---

### Decision · Program_Chairs · 2024-09-25

**Decision:**

Accept (poster)

**Comment:**

Nearly all reviewers recommend acceptance of this paper. The sole exception is reviewer EHt5, who provided valuable feedback and raised valid concerns. The authors have addressed some of these concerns, but additional experiments are necessary to fully resolve these issues. However, I believe the current set of experiments is sufficient to demonstrate the capabilities and some limitations of the model.

That being said, I agree with reviewer Eht5 that the current set of experiments does not fully support the claim of "SOTA performance in demanding generalization tasks." This claim should be rephrased to be less assertive. However, I don't think this is enough grounds for rejecting the paper. The paper provides valuable theoretical and empirical contributions to the field.

I encourage the authors to incorporate the results from the rebuttal, make the SOTA claim more moderate, and overall recommend this paper for acceptance.